# Diagnosing failures of fairness transfer across distribution shift in real-world medical settings

**Jessica Schrouff** *
Google Research
schrouff@google.com

**Natalie Harris**
Google Research

**Oluwasanmi Koyejo**
Google Research

**Ibrahim Alabdulmohsin**
Google Research

**Eva Schnider** †
University of Basel

**Krista Opsahl-Ong**
Google Research

**Alex Brown**
Google Research

**Subhrajit Roy**
Google Research

**Diana Mincu**
Google Research

**Christina Chen**
Google Research

**Awa Dieng**
Google Research

**Yuan Liu**
Google Research

**Vivek Natarajan**
Google Research

**Alan Karthikesalingam**
Google Research

**Katherine Heller**
Google Research

**Silvia Chiappa**
DeepMind

**Alexander D'Amour**
Google Research

## Abstract

Diagnosing and mitigating changes in model fairness under distribution shift is an important component of the safe deployment of machine learning in healthcare settings. Importantly, the success of any mitigation strategy strongly depends on the *structure* of the shift. Despite this, there has been little discussion of how to empirically assess the structure of a distribution shift that one is encountering in practice. In this work, we adopt a causal framing to motivate conditional independence tests as a key tool for characterizing distribution shifts. Using our approach in two medical applications, we show that this knowledge can help diagnose failures of fairness transfer, including cases where real-world shifts are more complex than is often assumed in the literature. Based on these results, we discuss potential remedies at each step of the machine learning pipeline.

## 1 Introduction

As progress is made to integrate machine learning (ML) systems in real-world applications, one concern is how the ML model's behaviour might be affected by changes in the data distribution. While extensive work has investigated the effect of distribution shift on ML model performance [see 40, 65, 17, 62, for reviews], recent work has also highlighted that fairness properties that are satisfied on the data used to develop a model might not always hold if the data distribution changes [28, 26]. In high-risk domains like healthcare, this can translate to 'if a model satisfies fairness criteria when trained in "Hospital A", it may not satisfy them when tested in "Hospital B"'. Fueled by this observation, a new field of research is emerging to understand the relationship between ML fairness

---

*Now at DeepMind

†Work performed while interning at Google Research

36th Conference on Neural Information Processing Systems (NeurIPS 2022).

and robustness to distribution shift [e.g. 50, 1, 60, 13, 72, 53] and to provide mitigation techniques for obtaining *robustly fair* ML models [e.g. 52, 55, 54, 73, 12].

Importantly, the success of robustly fair machine learning depends strongly on the nature of the distribution shifts that the system will encounter in practice. This point has been made elegantly in existing work as a motivation to use causal structure to provide fairness guarantees for specified distribution shifts [52, 55]. However, to the best of our knowledge, few methods can *diagnose* the nature of the shift and guide towards an appropriate mitigation strategy [54].

In this work, we design statistical tests that can, based on a simplified causal graph of an application and a small target dataset, assess the structure of the distribution shift encountered by the system and identify aspects of the data that hinder the transfer of fairness properties. Using applications in dermatology and Electronic Health Records (EHR), we show that this analysis is crucial for selecting an appropriate mitigation strategy. Further, our work also reveals that clinically plausible shifts are often more complex than can be handled by current mitigation techniques, highlighting the need for additional research into a broader set of remedies across the entire ML pipeline.

## 2   Background

**Notation**   We consider problems where we are given variables $X, A, Y$, where $X$ is a set of features, $A$ is one or more sensitive attributes, and $Y$ is an outcome or label to be predicted. Our goal is to build a classification or regression model $f(X')$, where, depending on the context, $X' := (X, A)$ or $X' := X$. We are concerned with the fairness properties of $f$, both on the training distribution and in one or more deployed environments. Depending on the setting, we focus on several statistical definitions of fairness [4]: gaps in subgroup accuracy, demographic parity, and equalized odds. See the Supplement for their formulation.

Formally, we represent distribution shifts using the so-called Joint Causal Inference (JCI) framework of Mooij et al. [35]. We represent causal relationships between elements of $A, X, Y$ in a causal Bayesian network (CBN, see the Supplement for an introduction). In a CBN, a node $U^i$ with an arrow into $U^j$ is called a *direct cause* of $U^j$. We call all direct causes of a node $U^i$ its causal parents, and denote them $\mathrm{pa}(U^i)$. To represent shifts, we augment the CBN with an environment variable $S$, where the event $S = 0$ indicates data from the source environment and $S = 1$ denotes data from the target environment, so that source data are distributed according to $P(A, X, Y \mid S = 0)$ and the target data to $P(A, X, Y \mid S = 1)$. This is called the *joint causal graph*. In the joint causal graph, arrows from $S$ denote changes to statistical relationships that are induced by the shift. Specifically, an arrow from $S$ to some variable $U$ indicates that the conditional distribution of that variable given its causal parents could change between training and deployment environments, while the absence of such an arrow would indicate that the conditional distribution remains the same, i.e. $P(U \mid \mathrm{pa}(U), S = 0) = P(U \mid \mathrm{pa}(U), S = 1)$.

**Components of Distribution Shifts**   In JCI formalism, the components of a distribution shift are represented by direct arrows from the shift variable $S$ to variables $A, Y$, and $X$. Here, we briefly summarize how these arrows can be interpreted. $S \to A$ arrows represent *demographic shifts*, implying $P(A \mid S = s)$ changes for different values of $s$. An example of such a shift is a difference in the age distribution of patients in $S = 0$ versus $S = 1$. $S \to X$ arrows represent *covariate shifts*[3], where the conditional distribution of input features $P(X \mid \mathrm{pa}(X), S = s)$ changes for different values of $s$. For example such a shift could result form using different cameras or views to acquire an image in $S = 0$ and $S = 1$ (c.f., acquisition shift in Castro and Glocker [5]). Finally, $S \to Y$ arrows represent *label shifts*[4], where the conditional distribution of labels $P(Y \mid \mathrm{pa}(Y), S = s)$ changes for different values of $s$. A label shift can occur through different disease prevalences, or different strategies for obtaining labels, across $S = 0$ and $S = 1$ (c.f., prevalence shift and annotation shift, respectively, in Castro and Glocker [5]).

In real-world applications, distribution shifts often include multiple components, where there are multiple arrows from $S$ to $\{A, X, Y\}$. We call such shifts *compound*. Compound shifts often occur when there is an unobserved variable that affects multiple variables whose distribution changes

---

[3]In the domain adaptation literature, "covariate shift" is often used to mean that *only* $P(X)$ changes. We call this "exclusive covariate shift".

[4]As above, we refer to the case where *only* $P(Y)$ changes as "exclusive label shift".

**Algorithm 1** (Conditional) independence testing assessing the nature of shift $S$ on a single variable $U \in \mathcal{G}$. $U$ represents the feature values or its summary if high-dimensional. $\odot$ is the Hadamard product.

---

**Require:** source dataset $\mathcal{D}$, target dataset $\mathcal{D}'$, a joint causal graph $\mathcal{G}$
    Split $\mathcal{D}$ into $\mathcal{D}_w, \mathcal{D}_t$; Split $\mathcal{D}'$ into $\mathcal{D}'_w, \mathcal{D}'_t$
    **for** $1{:}n_{bootstrap}$ **do**
        Sample $\mathcal{V}$ from $\mathcal{D}_t^{n_0 \times l}$ and $\mathcal{V}'$ from $\mathcal{D}'_t{}^{n_1 \times l}$
        Set $w_s = 1^{n_s}$ for $s \in \{0, 1\}$.

        **if** $\mathrm{pa}(U)$ `not` $\emptyset$ **then**                $\triangleright$ Compute balancing weights
            Sample dataset $\mathcal{Q} = \{\mathrm{pa}(U)|S = 0\}$ from $\mathcal{D}_w$ and $\mathcal{Q}' = \{\mathrm{pa}(U)|S = 1\}$ from $\mathcal{D}'_w$
            Build predictor $P(S|\mathrm{pa}(U))$ classifying $\mathcal{Q}$ from $\mathcal{Q}'$
            $w_s(\mathrm{pa}(U)) \propto P(\mathrm{pa}(U) \mid S = s)^{-1} \in \mathcal{R}^{n_s}$ for $s \in \{0, 1\}$.
        **end if**
        Compute t-statistic between $[w_0 \odot \mathcal{V}(U); -w_1 \odot \mathcal{V}'(U)]$ and $0$
    **end for**
    Compute p-value from t-test of bootstrapped t-statistics against $0$
    **return** p-value                    $\triangleright$ The p-value that $U \perp\!\!\!\perp S \mid \mathrm{pa}(U)$

---

between source and target. An example is that of a new device on the market [19]. Such a device could create qualitatively different images $X$, but it could also change the observed patient population by being made more available to certain subsets of demographic groups that tend to be more or less healthy. In this case, the shift induced by introducing the device simultaneously affects $A, X$ and $Y$.

**Shift Structure and Fairness Transfer**    The transfer of fairness depends on the causal structure of the application and the structure of the shift [52, 53]. In particular, implications for fairness transfer can be directly read off of conditional independences with respect to the shift $S$ in the joint causal graph. For example, for classifiers that take a set of variables $\{\mathbf{V}\}$ as input, the following metrics are known to be invariant across the shift under the following conditional independence conditions:

- Classification error ($P(Y \neq f(\mathbf{V}) \mid \mathbf{V}, S)$) is invariant if $Y \perp\!\!\!\perp S \mid \{\mathbf{V}\}$ [32, 52].
- Demographic parity gap ($\max_{a \in \mathcal{A}} E[f(\mathbf{V}) \mid S, A = a] - \min_{a \in \mathcal{A}} E[f(\mathbf{V}) \mid S, A = a]$) is invariant if $\{\mathbf{V}\} \perp\!\!\!\perp S \mid A$ [52].
- Equality of odds gap ($\max_{a,y \in \mathcal{A}\mathcal{Y}}[f(\mathbf{V}) \mid S, A = a, Y = y] - \min_{a,y \in \mathcal{A}\mathcal{Y}}[f(\mathbf{V}) \mid S, A = a, Y = y]$) is invariant if $\{\mathbf{V}\} \perp\!\!\!\perp S \mid Y, A$ [52].

In JCI formalism, the above conditions imply that fairness transfer requires certain arrows from $S$ to $\{A, X, Y\}$ to be missing. The nature of the shift also strongly influences which mitigation strategies are applicable. We discuss implications for mitigation strategies in more detail in Sec.5.

## 3   Empirical tests for assessing shift structure

Given the central role of distribution shift structure in fairness transfer, our main methodological proposal is to investigate the structure of shifts empirically when fairness fails to transfer. In this section, we outline a general strategy for testing which components of a shift are present, given data from source and target distributions, and a BCN for the application.

**Approach.** For each variable $U$ in the graph (i.e. each $A, X$ and $Y$), we assess whether there is a direct effect of the environment $S$ on $U$. We test the null hypothesis $H_0$: $P(U \mid \mathrm{pa}(U), S = 0) = P(U \mid \mathrm{pa}(U), S = 1)$. Rejecting this hypothesis implies an arrow $S \to U$ in the joint causal graph. If the marginal distribution of $U$ differs across source and target distributions, this comparison can be challenging. To isolate the direct effect of $S$, we compute weights that balance the causal parents $\mathrm{pa}(U)$ across environments [44, 23]. We then test the hypothesis that the reweighted distributions of $U$ match across the environments $S$. For simplicity, we test whether the reweighted means are equal, and in cases where the dimensionality of $U$ is high, we conduct this test on functions of $U$ that reduce the dimensionality, following strategies in Rabanser et al. [43]. [5] This procedure is described in Algorithm 1 and detailed in the Supplement.

---

[5] A rejection of these simplified hypotheses implies that the original null hypothesis should also be rejected.

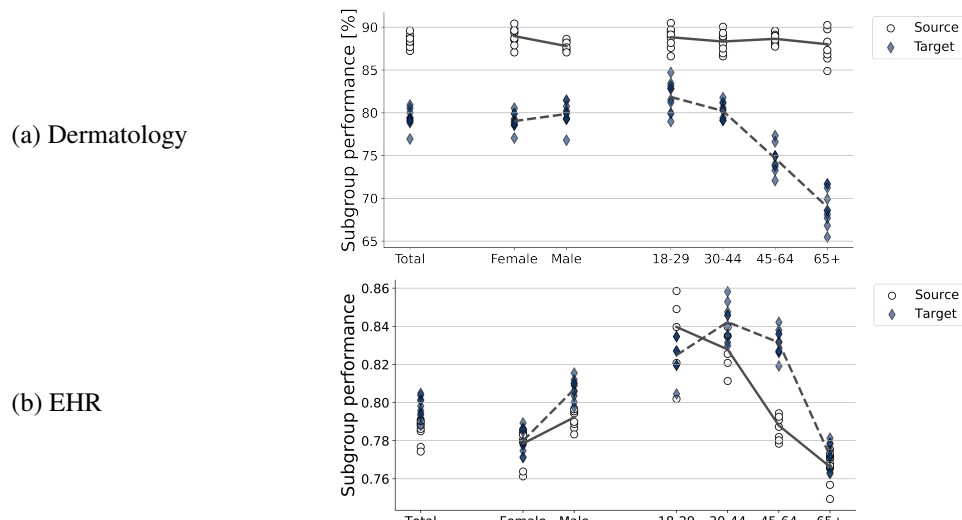

Figure 1: Model performance across subgroups (age and sex) on the source (circles with plain line) and target (diamonds with dashed line). Each marker represents one replicate of the model. (a) Top-3 accuracy (in %) in the dermatology application. (b) Accuracy in EHR.

**Validation.** We conduct three validation experiments for our testing procedure using data from our dermatology application in the Supplement. We first assess Type I error in an experiment where we test random splits of the same data against each other. Our test displays a false positive rate similar to our threshold of $5\%$ for hypothesis testing. We however note that the variance of this result increases with smaller numbers of samples. We then confirm non-trivial power in an experiment where we introduce a shift by subsampling younger patients with a particular skin condition in the source dataset (a shift on $A$ and $Y$). Our testing approach correctly identifies the dimensions of $A$ and $Y$ that were affected by $S$ due to the engineered shift. We replicate this test for a shift in $Y$ only, with the same result.

## 4 Distribution shifts in real-world healthcare applications

Here, we present two case studies from the healthcare domain in dermatology and in clinical risk prediction using Electronic Health Records in which fairness does not transfer (Fig. 1). As per [27], we consider the setting of 'dataset shift', whereby a model is developed on the source data and tested on the target data[6], which is a common setting in medical applications [66, 71, 27].

### 4.1 Predicting skin conditions in dermatology

In this task, we predict common skin conditions (26 conditions plus an additional 'other' category to capture the long tail), based on one or multiple images of the pathology of interest and additional metadata [31]. The data used as source is a subset of the data described in Liu et al. [31]. Briefly, it consists of de-identified retrospective adult cases from a teledermatology service serving 17 sites from 2 states in the United States. Each case contained 1-6 clinical photographs of the affected skin areas as well as patient demographic information and medical history (see Table S1 of [31]). We split the source data such that it comprises 12,024 cases for training, 1,925 for validation and hyper-parameter tuning and 1,924 for testing.

We consider how the fairness properties of a model trained in this setting transfer to a target dataset, unobserved at model development time, consisting of 1,843 labelled teledermatology cases from Colombia. In the Supplement, we consider transfer to a second target from skin cancer clinics in Australia, as well as the properties of a model jointly trained on the source and the second target, and deployed on the first target data. The model we consider is a ResNet architecture, as in [31, 46].

---

[6]We discuss fine-tuning and joint training in Sec.5, but this is not the setup of our work

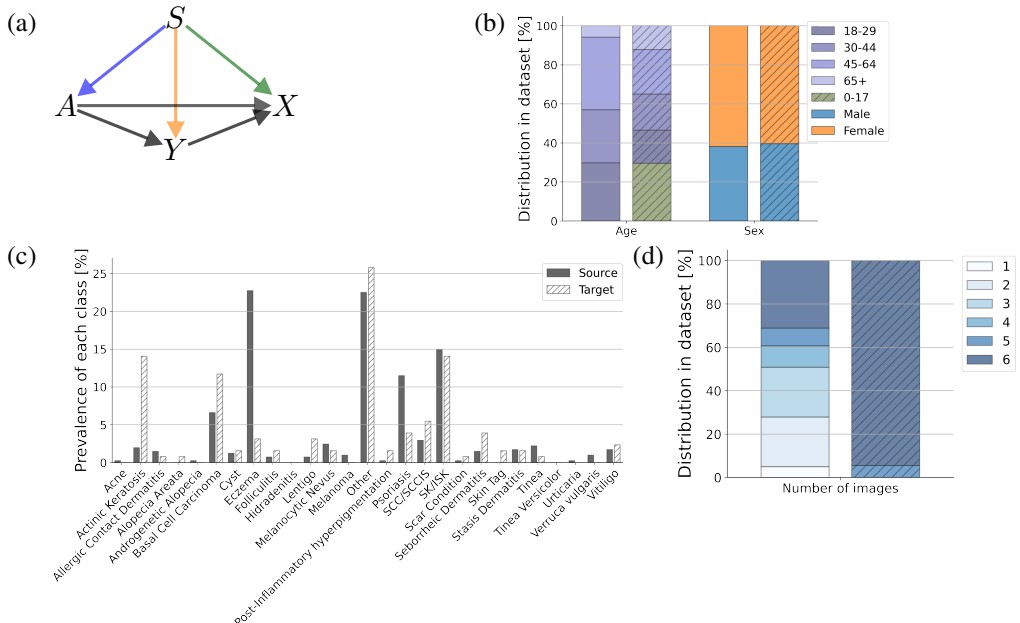

Figure 2: A compound shift in dermatology. (a) Simplified causal structure of the application (see text for variable description). Colored arrows represent statistical dependence as identified by Algorithm 1. (b) Distribution in the source and in the target dataset of the sensitive attributes, computed in terms of percentage of pre-defined subgroups. (c) Prevalence of each condition for female patients over 65 years of age. (d) Distribution of the number of images in cases labelled as 'SK/ISK' in older females. Each case includes min. 1, max. 6 images whose embeddings are then averaged. The distributions in the source data are presented on the left, and their corresponding distributions in the target data on their immediate right with a hashed pattern.

We assess model performance and fairness across 10 replicates of the model on the source and on the target (Fig. 1). On the source data, the model performs similarly across all subgroups and could reasonably be assessed as 'fair'. However, the maximum gap between age groups increases starkly on the target (from $1.05\%$ to $12.86\%$).

To understand the failure of fairness transfer in this context, we investigate the structure of the shift by referring to Algorithm 1. We begin by drawing a simplified causal graph $\mathcal{G}$ of the dermatology task at hand (Fig.2(a)). We consider this an anti-causal learning task [49], in which the skin condition label $Y$ is a cause of the input image $X$, and consider demographic information $A$ to be a causal parent of both $X$ and $Y$. We omit variables that are unobserved in either the source or target data. [7]

We now test whether the shift $S$ has direct effects on $A$, $Y$, or $X$. Overall, we find that the shift is compound, and affects all three aspects of the data distribution. We begin with $A$, which comprises age and sex.[8] Because $A$ has no causal parents in our graph (besides, potentially, $S$) we directly assess differences in age and sex across environments using classical tests. Age has a different distribution across the two datasets (see Fig. 2(b)), with the population in the source data being typically younger (median: 40 years old, $25\%$ quantile: 27, $75\%$ quantile: 54) than in the target data (median: 44 years old, $25\%$ quantile: 30, $75\%$ quantile: 59). Sex distributions are well matched across the source and target. These results suggest a direct effect of $S$ on $A$, and we add this relationship to the causal graph (Fig. 2(a), in purple).

We then assess whether $S$ directly affects the labels $Y$. In our graph, $A$ is a causal parent of $Y$, so following our testing strategy in Algorithm 1, we test whether $P(Y \mid A, S = 0) = P(Y \mid A, S = 1)$. (Recall that conditioning on $A$ isolates the direct effect of $S$ on $Y$.) Using 27 univariate tests (one per condition, [43]), we obtain significant differences between $P(Y \mid A, S = 0)$ and $P(Y \mid A, S = 1)$ for 3 conditions ($p < 0.05$, Bonferroni corrected). Fig.2(c) illustrates the label shift in a specific age and sex subgroup (here females aged 65+, 409 cases in the source, 128 in the target data). We see

---

[7]The metadata available in the source data includes co-morbidities/medical history and symptoms, but these are absent in the target datasets.

[8]Sex is mainly recorded by clinicians in the source. Sex is self-reported in the target.

that the source data includes more cases of 'eczema' and 'psoriasis' while conditions like 'actinic keratosis' and 'seborrheic dermatitis' are more represented in the target. These results suggest that the environment also directly affects the labels (orange link in Fig.2(a)).

Finally, we consider whether the features of the images themselves (designated by $X$) are directly affected by $S$. Conditioning on causal parents of $X$, we test whether $P(X \mid Y, A, S = 0) = P(X \mid Y, A, S = 1)$. We summarize $X$ by training a model $f(X) \to Y$ on the source, a strategy proposed in Rabanser et al. [43]. Our weighted tests suggest a significant difference between the two distributions ($p < 0.05$ corrected on 1 dimension). Fig.2(d) illustrates this difference by computing the number of images per case in the group of older females considered above with cases labelled as 'SK/ISK' ( source median: 3, $n = 61$, target median: 5, $n = 18$). This result suggests the existence of the direct path $S \to X$ and we add this relationship to the graph (green link in Fig.2(a)).

Based on these results, we observe that all aspects of the data are affected by the environment. Thus, none of the conditional independence conditions for fairness transfer hold in this setting. However, we also see that some conditions might display invariance once controlled for demographics, which suggests possible mitigations that we explore in Section 5. The Supplement includes experimental details and further results on the second target and the joint training.

## 4.2 Clinical risk prediction from Electronic Health Records

In this application, we predict the clinical outcomes of patients based on EHRs. EHRs record the time series of interactions between a patient and the clinical system (e.g. medication, labs, vitals, . . . ). Distribution shift is a major concern for systems that consume EHR data, which can be heterogeneous across centers [67, 53], and even within a single system [37]. As an example, over half of hospitals in the US with intensive care units (ICUs) only have a single unit where all critically ill patients (medical, surgical, cardiac, etc.) are admitted [9]. Risk scores, such as the Acute Physiology and Chronic Health Evaluation II (APACHE II) score are used to evaluate all critically ill patients, but do not perform equally across clinical subgroups, e.g. underperforming on patients with cardiac disease [42]. It would be desirable to avoid such issues with risk scores constructed through machine learning. We use the open access, de-identified Medical Information Mart for Intensive Care III (MIMIC-III) dataset [24], which consists of data from admissions to ICUs at the Beth Israel Deaconess Medical Center between 2001 and 2012. This clinical system has the benefit of having separate specialized ICUs and allows to assess the generalizability of risk scores estimated in one clinical population to another. Based on clinical input, we consider the Medical ICU (MICU), Surgical ICU (SICU) and Trauma Surgical ICU (TSICU) to be generalist ICUs (source data); whereas the Cardiac Surgery Recovery Unit (CSRU) and Coronary Care Unit (CCU) are specialized ICUs (target data). This split leads to 17,641 patients included in the source dataset and 10,442 patients in the target dataset, after selecting adult patients with a length of stay of minimum one day and with a recorded care unit. Our goal is to obtain a robustly fair model that predicts prolonged ICU stay (i.e. length of stay > 3 days, as in [63, 41]) using data from the first 24 hours of first ICU admission and a recurrent architecture [57, 58, 47]. The model is trained on 80% of the source data, tuned using a separate split of 10% and tested on the remaining 10%. See the Supplement for further details. Noteworthy, the model does not have access to the 'reason for visit'.

Figure 1(b) shows that the fairness properties of this model, as measured by gaps in subgroup performance, demographic parity (DP) and equalized odds (EO), do not transfer across ICU types. The model displays unfairness w.r.t. age in both datasets (maximum gap $\sim 7\%$, similar DP and EO scores). However, the pattern of unfairness is different in the two datasets, with e.g. the 45-64 years old subgroup being under-served in the source but over-served in the target. The performance gap between sex groups increases from 1% in the source (DP: $0.002\pm0.002$) to 2.7% (DP: $0.016\pm0.003$) on the target.

To understand this failure of fairness transfer, we examine the structure of the shift. First, we draw a causal graph for the variables in this problem, building on the work in Singh et al. [52]. Here, we incorporate demographics $A$ as age and self-reported sex, comorbidities $M$ as previous medical history defined by ICD-9 codes [7, 14, 22, 29], observed features $X$ as all labs and vitals, and $Y$ as our prolonged length of stay label. As a more comprehensive feature set leads to better predictive

---

[9]Critical Care Statistics. `https://www.sccm.org/Communications/Critical-Care-Statistics`

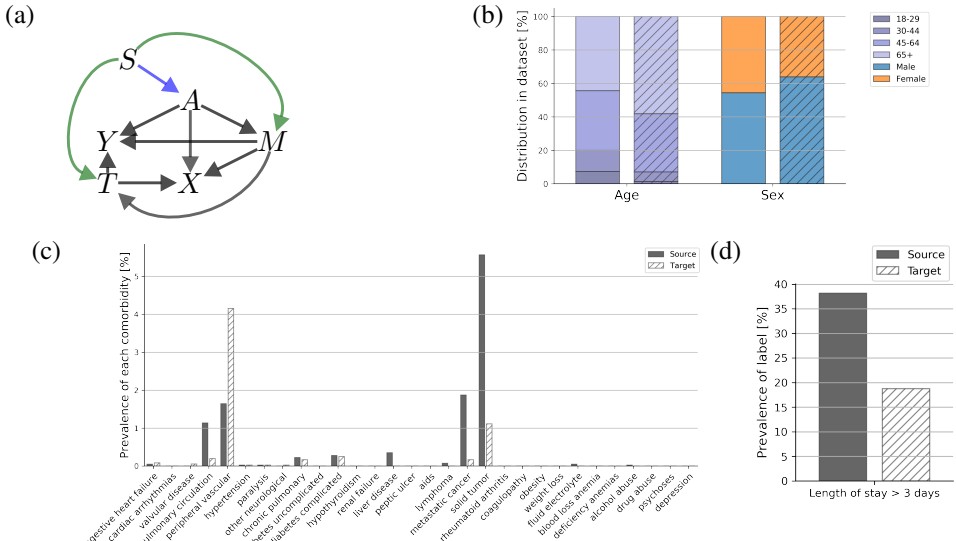

Figure 3: A compound shift in EHR. (a) Causal graph for this application (see text for variable description). Colored arrows represent statistical dependence as identified by Algorithm 1. (b) Distribution of the sensitive attributes, computed in terms of percentage of pre-defined subgroups. (c) Prevalence of comorbidities for male patients over 65 years of age. (d) Prevalence of the label for male patients over 65 with a solid tumor comorbidity and under vasopressors/inotropes. The distributions in the source are represented on the left, and their corresponding distributions in the target on their immediate right with a hashed pattern.

performance [47], we use all 59,351 features at our disposal and add 'treatments' $T$ (e.g. medications) to the graph (Fig. 3(a)). In our graph, $S$ represents the ICU unit type that a patient is admitted to.

We now test whether the shift $S$ has direct effects on the variables in this problem using our proposed approach. Beginning with $A$, which has no causal parents, we test whether $P(A \mid S = 0) = P(A \mid S = 1)$. We estimate the distribution of age and sex in each dataset (Fig. 3(b)). We observe that the population in the source is typically younger than the population in the target (t-test: $p < 0.0001$). In addition, the proportion of males in the source and target differ (source: 50%, target: 65%). Therefore, $S \rightarrow A$.

To assess the shift's effect on comorbidities $M$, we refer to Elixhauser and Vanwalraven scores [16, 59] computed on the source and target data [25][10]. Figure 3(c) displays the distribution of the 30 comorbidities for older males ($n = 3,951$ in source and $n = 3,588$ in target). As expected, even after adjusting for the causal parent $A$, patients in the target data have a higher prevalence of comorbidities related to the peripheral vascular system than in the source data and $M \not\perp\!\!\!\perp S \mid A$ for 5 comorbidities (weighted tests: $p < 0.05$, Bonferroni corrected).

We repeat the analysis to assess the direct effect of $S$ on the treatments ($P(T \mid A, M, S = 0) = P(T \mid A, M, S = 1)$ : $p < 0.05$), on the labs and vitals ($P(X \mid A, M, T, S = 0) = P(X \mid A, M, T, S = 1)$ : $p > 0.05$) and on the label ($P(Y \mid A, M, T, S = 0) = P(Y \mid A, M, T, S = 1)$ : $p > 0.05$) in the Supplement. Figure 3(d) displays the prevalence of prolonged length of stay for older males with a solid tumor comorbidity and receiving vasopressors/inotropes ($n = 55$ in source and $n = 16$ in target).

Our results suggest that the shift is compound, i.e. the ICU unit a patient is admitted to changes the relationships between multiple variables in our causal graph. We note that this conclusion is not surprising clinically, given that the environment $S$ is correlated with the reason for their ICU admission, and that this reason for admission (unobserved) is a main driver for all variables. As in the dermatology case, the compound shift leads to multiple paths for $S$ and $A$ to affect the covariates $M$ and $X$ and the label $Y$, explaining the failure of fairness transfer.

---

[10]Code publicly available at `https://doi.org/10.5281/zenodo.821872`

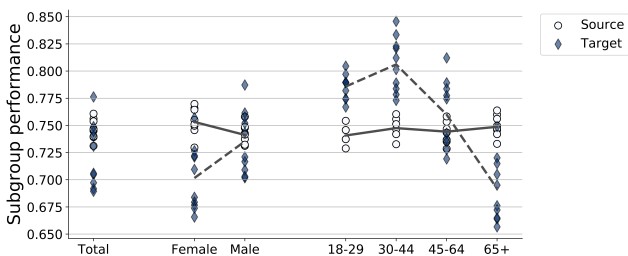

Figure 4: EHR model accuracy across subgroups after post-processing of the predictions for equalized odds based on age.

## 5 Mitigation and related work

A variety of fairness, robustness and robustly fair mitigation strategies have been proposed. We contextualize a number of methods in terms of anti-causal and causal learning tasks [49] in the Supplement, and highlight how understanding the nature of the distribution shift can guide mitigation.

**Fairness mitigation.** Pre-, in- and post-processing techniques for mitigating unfairness have been proposed [see 4, 38, for overviews]. Fairness metrics and mitigation techniques currently used in real-world applications are typically evaluated on a single learning task or data distribution [e.g. 74, 2, 64, 41, 9]. More similar to our settings, the work by Creager et al. [12] addresses fairness in dynamical systems, although only for low-dimensionality variables. Based on our causal framing, applying fairness mitigations in the source domain could potentially aid in fairness transfer by cutting some edges in the underlying CBN [60]. However, fairness mitigation on the source only leads to provable fairness transfer in specific settings, i.e. mostly when the shift is demographic (Supplement).

Based on this analysis, performing fairness mitigation in the presence of a compound shift would fail to provide a fair result on the target. Considering the applications above, a reasonable option would be to perform fairness mitigation for the EHR task based on age. We perform post-processing of the predictions to enforce either demographic parity or equalized odds across age subgroup using the method in Alabdulmohsin and Lučić [3]. This results in a decrease in demographic parity to value $10^{-5}$ in the source, but to (slightly) increase to value $0.078 \pm 0.025$ in the target data. Similarly, equalized odds decrease to value $10^{-5}$ and the gap in model accuracy between groups is reduced from 7% to 0.8% (Fig. 4) in the source, but increases from $0.05 \pm 0.008$ to $0.16 \pm 0.04$ on the target data, and the gap between age groups increases to 11.4%. We therefore observe that mitigating for either demographic parity or equalized odds on the source leads to no improvement, or even worsening of the fairness properties on the target, as predicted by our analysis.

**Robustness mitigation:** Relations between robustness guarantees and individual and group fairness have been studied in [70, 36, 68, 69] and [60] respectively. [1, 13, 72] have investigated the relationship between fairness and distribution shift experimentally by studying how robustness mitigation strategies affect fairness metrics. These works do not provide provable robustly fair models.

**Robustly fair mitigation.** Work explicitly investigating fairness properties under distribution shift can mostly be divided into methods that provide a priori *guarantees* or *bounds* [e.g. 50, 52, 55] and methods that *adapt* to the new distribution [e.g. 11, 39, 54, 73, 26, 15, 56].

Recent work by Singh et al. [52] suggests a straightforward mitigation with guarantees, based on selecting features that satisfy the fairness transfer criteria outlined in Section 2. However, this approach is limited under compound shifts with complex data. Most notably, it cannot be used in cases where there is a label shift ($S \to Y$). However, even when there is no label shift, under compound shifts the criteria for feature selection can yield a trivial predictor where no observed variables can be used. This is the case in our EHR example under the joint causal graph in Fig. 3(a). We discuss the applicability of [52] across different causal scenarios in the Supplement. In addition, feature selection approaches are more suited for semantic features and, while it is technically feasible, would not lead to insightful results in non-semantic features such as pixels. In computer vision, masking has been used previously as an approximation to feature selection to constrain the learning towards specific areas of the image [45, 51]. While this technique has shown some promise with chest x-rays [61], it is impractical due to the additional requirement of a mask for all training images.

In addition, it increases the ratio of signal of interest compared to undesirable signal, but does not exclude such signals and comes with no guarantee.

Another line of work considers adapting transfer learning and meta-learning algorithms to consider fairness constraints and therefore ensure generalization to new tasks also in terms of fairness [11, 39, 54, 73]. Specifically, Coston et al. [11] modify the standard approach to transfer learning under covariate shift by adding a constraint to the weight learning objective that enforces closeness between all pairs of protected groups. Schumann et al. [50] provide bounds on fairness transfer by adding adversarial heads on both the in-source attribute $A$ and the environment $S$. When we cast this work into our causal framework, we notice that it would provide a robustly fair model under an exclusive covariate shift in an anti-causal learning task. Slack et al. [54] propose meta-learning with a regularizer that imposes demographic parity or equalized odds to improve the transfer of fairness. The method requires a small set of labelled target data (K-shot learning) and multiple tasks, and comes with no guarantees. Work on fine-tuning and transfer learning [39, 73] leverages task similarity to learn fair representations that provably generalize to unseen tasks, similarly requiring a small set of labelled target data. Importantly, these methods generate one tuned model per environment, which is not suitable for all real-world deployments of ML. It is also unclear whether these techniques could handle more complex tasks, e.g. predictions from sparse multivariate timeseries as in the case of EHR.

**Shift detection.** Many methods exist to detect covariate shift [e.g. 33, 6] or label shift [30, 20]. They however often rely on the assumption that other variables are not affected by the shift. For instance, Lipton et al. [30] relies on the assumption that $X \perp\!\!\!\perp S \,|\, Y$. On the other hand, Slack et al. [54] introduce a technique to identify mean shifts that could lead to unfairness. While the method is an interesting step towards 'fairness transfer warnings', this strategy does not allow to guide the selection of mitigation. Recently, Singh et al. [53] investigate differences in source and target distributions using squared maximum mean discrepancy [21], but do not take the causal structure into account. To differentiate direct from indirect effects of $S$ on variables $U$, they perform causal inference and visually inspect the obtained graph. This method can only be applied to low-dimensionality datasets.

# 6   Discussion

In this work, we applied statistical tests within a joint causal framework [35] to better understand failures of fairness transfer across distribution shift. Our approach tests for direct effects of the environment on every random variable in a task. We use this technique to characterize distribution shifts that result in two real world failures of fairness transfer in the healthcare domain. We show that this knowledge can then guide the choice of an appropriate mitigation strategy, when applicable.

Our work highlights real-world challenges that prevent the development of robustly fair models. Given that each task leads to different complexities in terms of causal graph, expected shifts, and fairness requirements, these findings support looking beyond purely algorithmic solutions. Considering the whole ML pipeline, we can re-interpret the remedies discussed in Chen et al. [8] for fairness transfer:

1. **Problem selection.** Consider focusing on tasks with lower anticipated harms or clinical/policy safeguards against ML unfairness.
2. **Data collection.** Consider early data collection from anticipated deployment environments in which distribution shifts are suspected. This would allow to perform (conditional) independence tests to guide further mitigation.
3. **Outcome definition.** Consider intermediate outcomes for which unfairness under distribution shift might be less consequential (e.g. image segmentation compared to diagnosis).
4. **Algorithm development.** Based on (2), some strategies can be used if their assumptions are satisfied. In addition, in the present work we used a causal framework. However, techniques that are not easily cast into this framework exist. For instance, manual feature engineering has been used to improve the robustness of EHR models to distribution shift [37]. While this work has not evaluated the fairness of the model in the source and target distributions, we believe that feature engineering might be a promising avenue to enforce specific inductive biases during learning while reducing the risks of distribution shifts. We however note that our work does not assess the impact of model architecture or different training strategies, which might lead to compounded bias and/or poor generalization.

5. **Post-deployment.** Further safeguards may be envisaged to safely deploy ML systems, e.g. with a prospective observational (non-interventional) integration [19]. Transfer learning or retraining with site-specific updates might then be considered as necessary prior to interventional deployment. Finally, those deployments may be accompanied with robust post-market surveillance to continuously monitor fairness.

**Limitations and future work.** The power of our diagnosis method depends on the number and quality of the samples. Therefore, it should be perceived as displaying potentially problematic aspects of the data, rather than assessing invariant features. This is a common limitation in shift detection [30] and mitigation [52]. Furthermore, the data at hand usually represents a set of proxies for the underlying variables. Therefore, even if a shift seems to satisfy some invariance assumptions, there are no sharp guarantees [18]. This is especially true for demographic factors: features such as ethnicity, age or sex only approximate sensitive attributes. We are also limited by which factors are observed across all environments and cannot assess changes in unobserved variables (e.g. social determinants of health).

While our work covers different types of shift, future work should investigate further sources of shift and biases, such as sampling biases or missingness [5, 72]. In addition, extending the testing strategy to compare multiple (discrete or continuous) versions of the environment $S$ would be valuable.

By specifying causal graphs for each problem, we make assumptions about the relationships between variables. These assumptions might not represent the true underlying data generation process. In addition, the effects of $A$ could potentially be divided into 'fair' and 'unfair' effects, as proposed in [10, 34]. For real-world applications, further assumptions were made, e.g. considering the silver standard label in dermatology as a good approximation for the gold standard label (hence an anti-causal task) [5]. The value of these graphs is purely illustrative and should not be considered for medical applications without validation.

In terms of metrics, we have focused mostly on demographic parity and equalized odds, due to their recent causal grounding [60]. Different metrics might, however, be affected differently, and it is possible that some metrics are more or less 'sensitive' to distribution shift. Similarly, different fairness mitigation strategies might be affected differently by distribution shift, although none provides guarantees under distribution shift. We also note that 'fairness' in the healthcare domain is an active discussion, and that equalized odds or demographic parity metrics do not consider factors such as social determinants of health or health equity. As discussed in [41], considering human elicited metrics might be an avenue forward. Similarly, 'fairness' might change depending on the context [48] and different metrics might be relevant in different environments.

## Acknowledgments and Disclosure of Funding

The authors would like to acknowledge and thank Lucas Dixon, Noah Broestl, Sara Mahdavi, Nenad Tomasev, Cameron Chen, Stephen Pfohl, Matt Kusner, Victor Veitch, Jon Deaton, Shannon Sequeira, Abhijit Guha Roy, Jan Freyberg, Aaron Loh, Martin Seneviratne, Patricia MacWilliams, Yun Liu, Christopher Semturs, Dale Webster, Greg Corrado and Marian Croak for their contributions to this effort. This work was funded by Google.

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
