# Diagnosing failures of fairness transfer across distribution shift in real-world medical settings: Supplement

**Jessica Schrouff** *
Google Research
schrouff@google.com

**Natalie Harris**
Google Research

**Oluwasanmi Koyejo**
Google Research

**Ibrahim Alabdulmohsin**
Google Research

**Eva Schnider** †
University of Basel

**Krista Opsahl-Ong**
Google Research

**Alex Brown**
Google Research

**Subhrajit Roy**
Google Research

**Diana Mincu**
Google Research

**Christina Chen**
Google Research

**Awa Dieng**
Google Research

**Yuan Liu**
Google Research

**Vivek Natarajan**
Google Research

**Alan Karthikesalingam**
Google Research

**Katherine Heller**
Google Research

**Silvia Chiappa**
DeepMind

**Alexander D'Amour**
Google Research

## Abstract

This document includes supplementary materials for the healthcare experiments from the manuscript titled *Diagnosing failures of fairness transfer across distribution shift in real-world medical settings*.

## 1 Methods

In this section, we provide further details on the methods, including the mathematical definitions of our metrics and an in-depth description of our testing procedure.

### 1.1 Fairness metrics

Our work focuses on statistical group definitions of fairness [3]. In particular, we frame the discussion around demographic parity in which the model's output is statistically independent of the sensitive attribute, i.e. $f(X') \perp\!\!\!\perp A$, and equalized odds in which the independence holds conditionally on the outcome, i.e. $f(X') \perp\!\!\!\perp A \,|\, Y$. In the experiments, these metrics are computed for binary tasks. We also compute the gap in subgroup accuracy. For all fairness metrics, the lower, the better.

We define demographic parity for a predictor $f$ as [8, 39, 18, 1]:

$$\max_{a \in \mathcal{A}} \mathbb{E}_{\mathbf{x}}[f(\mathbf{x}) \,|\, A = a] - \min_{a \in \mathcal{A}} \mathbb{E}_{\mathbf{x}}[f(\mathbf{x}) \,|\, A = a] \tag{1}$$

---

*Now at DeepMind

†Work performed while interning at Google Research

Equalized odds is computed in a similar fashion by conditioning on the positive and negative classes, taking the average of the discrepancies across classes.

Subgroup performance is computed as the accuracy (top-1 or top-3 in the case of dermatology, accuracy for EHR) within each subgroup. When the sensitive attribute is non-binary, we compute the maximum difference between any pair of subgroups.

## 1.2 Causal framework

A *Bayesian network* [6, 14, 20, 21] is a *directed acyclic graph* (DAG) $\mathcal{G}$ whose nodes $X^1, \ldots, X^D$ represent random variables and links express statistical dependencies among them. Each node $X^d$ is associated with a *conditional probability distribution* (CPD) $p(X^d \mid \mathrm{pa}(X^d))$, where $\mathrm{pa}(X^d)$ denote the *parents* of $X^d$, namely the nodes with a link into $X^d$. The joint distribution of all nodes is given by the product of all CPDs, i.e. $p(X^1, \ldots, X^D \mid \mathcal{G}) = \prod_{d=1}^{D} p(X^d \mid \mathrm{pa}(X^d))$. This function is assumed to be invariant across distributions. A *path* from $X_i$ to $X_j$ is a sequence of linked nodes starting at $X_i$ and ending at $X_j$. A path is called *directed* if the links point from preceding towards following nodes in the sequence. A node $X_i$ is an *ancestor* of a node $X_j$ if there exists a directed path from $X_i$ to $X_j$. In this case, $X_j$ is a *descendant* of $X_i$. We say that a set of nodes $\{X^i, \ldots, X^j\}$ *d-separates* two random variables $U$ and $W$ if $U \perp\!\!\!\perp W \mid \{X^i, \ldots, X^j\}$. A causal Bayesian network is a Bayesian network in which a link expresses causal influence rather than statistical dependence. In causal Bayesian networks, directed paths are called *causal paths*.

## 1.3 Assessing the causal structure of shifts

### 1.3.1 General Strategy

To assess the causal structure of a shift, we examine whether there is a direct effect of the shift $S$ on a focal variable $U$. This requires controlling for all other pathways by which $U$ may depend on $S$. To do this, we select a set of variables $\mathbf{V}$ such that $\{\mathbf{V}\}$ blocks all indirect paths from $S$ to $U$. We then test the following equality in conditional distributions: $P(U \mid \mathbf{V}, S = 0) = P(U \mid \mathbf{V}, S = 1)$ almost everywhere.

To test the equality of conditional distributions, we reduce the problem to testing whether the means of the two marginal distributions of $U$ are equal when the distributions of the background variables $\mathbf{V}$ are adjusted to follow the same distribution. Using observed data from environments $S = 0$ and $S = 1$, we can perform such an apples-to-apples comparison using importance weighting. In particular, for any distribution $\pi(\mathbf{V})$,

$$P(U|\mathbf{V}, S = 0) = P(U|\mathbf{V}, S = 1) \quad \text{for almost all } \mathbf{V} \quad \implies \quad (2)$$

$$\int E[U \mid \mathbf{V} \mid S = 0]\pi(\mathbf{V})d\mathbf{V} = \int E[U \mid \mathbf{V} \mid S = 1]\pi(\mathbf{V})d\mathbf{V} \quad \implies \quad (3)$$

$$E\left[\frac{\pi(\mathbf{V})}{P(\mathbf{V} \mid S = 0)}E[U \mid \mathbf{V} \mid S = 0] \mid S = 0\right] = E\left[\frac{\pi(\mathbf{V})}{P(\mathbf{V} \mid S = 1)}E[U \mid \mathbf{V} \mid S = 1] \mid S = 1\right] \implies$$
$$(4)$$

$$E\left[\frac{\pi(\mathbf{V})}{P(\mathbf{V} \mid S = 0)}U \mid S = 0\right] = E\left[\frac{\pi(\mathbf{V})}{P(\mathbf{V} \mid S = 1)}U \mid S = 1\right] \quad (5)$$

Define the weights $w_0(\mathbf{V}) := \frac{\pi(\mathbf{V})}{P(\mathbf{V}|S=0)}$, and $w_1(\mathbf{V}) := \frac{\pi(\mathbf{V})}{P(\mathbf{V}|S=1)}$. This result shows that we can test the equality of the conditional distributions by testing the implication that the mean of weighted outcomes is the same.

This leaves a degree of freedom for choosing the test distribution $\pi(\mathbf{V})$ on which the conditional distributions will be compared. This distribution then determines the weighting scheme that will be used. We define $\pi(\mathbf{V})$ to be a constant, i.e., uniform. Then the weights are

$$w_0(\mathbf{V}) \propto P(\mathbf{V} \mid S = 0)^{-1} \quad \text{and} \quad w_1(\mathbf{V}) \propto P(\mathbf{V} \mid S = 1)^{-1}.$$

This weighting scheme is known as inverse probability weighting or IPW [26, 11], and is a popular choice. We however note that other weighting schemes could be considered, e.g. overlap weights [15] or permutation weighting [2].

In practice, the likelihoods $P(\mathbf{V} \mid S = s)$ used to define the weights need to be learned from data. Usefully, because of how the weights are normalized, the likelihoods can be replaced with classification scores that estimate $P(S = s \mid \mathbf{V})$, leading to estimated weights $\hat{w}_0$ and $\hat{w}_1$. The choice of the classifier can be informed by the expected relationships between $\{\mathbf{V}\}$ and $S$: for instance, logistic regression can be considered in the case of simple relationships, while more expressive classifiers such as gradient boosted trees can be considered in the case of non-linear relationships [2]. As discussed in Arbour et al. [2], all classifiers are tuned (C for logistic regression, number of trees, tree depth and learning rate for boosted trees) in a cross-validation setup.

We then use a standard t-test to test the null hypothesis

$$H_0 : E[\hat{w}_0 U \mid S = 0] - E[\hat{w}_1 U \mid S = 1] = 0.$$

To decrease the variance in the obtained t-statistics, we bootstrap the weighting and testing procedure. The final p-value associated with $H_0 : P(U \mid \mathbf{V}, S = 0) = P(U \mid \mathbf{V}, S = 1)$ is the p-value of a two-sided $\mathcal{Z}$-test on the t-statistics obtained from each bootstrap.

**Observations and tips** From our observations, we note that it is best to allocate the largest available sample to estimate the likelihood ratio, compared to performing the final statistical test. We also observed that gradient boosted trees can lead to extreme weights in some cases. This results in an increased variance and non-significant results. We therefore clipped the weights to a maximum of 10 (arbitrary choice) to mitigate this effect.

### 1.3.2 Testing with High-Dimensional $U$

If $U$ has low dimensionality, Rabanser et al. [24] show that multiple one-dimensional tests can be used with correction for multiple comparisons (here Bonferroni). On the other hand, if $U$ has higher dimensionality (e.g. an image), summaries of $U$ can be constructed and the test can be performed on these lower-dimensional summaries. The validity of this approach follows from the fact that if the conditional distributions of $U$ are the same, then so will the conditional distributions of any summary $f(U)$. The trade-off is that the test loses all power to detect distributional differences that are compressed out by the summary $f(U)$, or to highlight specific dimensions in $U$ that are more or less affected by $S$. To define a summary that is relevant to the problem, Lipton et al. [16] suggest defining $f(U)$ to be the output of a model the predicts some variable of interest (say the outcome $Y$) using $U$. As reported in Rabanser et al. [24], other summarizing techniques could be considered.

### 1.3.3 Sanity checks with engineered shifts

We assess the specificity and sensitivity of our testing procedure (see Algorithm 1) using engineered shifts on the dermatology data. To estimate Type I error, we compare the source data to itself, which should lead to all variables being independent of the environment. More specifically, we select $10,000$ random samples from the dataset to compute the weights, and another $1,000$ random samples to apply the weights on and perform the weighted test. These sets are boostrapped 100 times (separately for the source and target sets) and we assess the proportion of false positives that is obtained for each variable $U$. We expect that we will obtain $5\%$ of false positives, as defined by our hypothesis testing threshold. As we do not expect any relationship between $\{\mathbf{V}\}$ and $S$, we use a logistic regression to estimate $P(S = s \mid \mathbf{V})$.

When comparing $P(Y \mid A, S = 0)$ to $P(Y \mid A, S = 1)$, we observe that the accuracy of the classifier mirrors the chance level of $50\%$ and that the proportion of false positives varies between 2 and $8.5\%$ (Figure 1(a)). Similarly, comparing $P(X \mid Y, A, S = 0)$ to $P(X \mid Y, A, S = 1)$ when using a model $f(X) \rightarrow Y$ to summarize $X$ leads to a proportion of false positives between 2 and $7\%$ (Figure 1(b)).

As the number of samples used to compute the weights is an important component of our testing approach, we repeat the process while varying the number of samples from 100 to 5,000. Figure 1(c) displays that the variance on Type I error (here across conditions) decreases when the number of samples to fit the $P(S = s \mid \mathbf{V})$ classifier increases.

To assess sensitivity, we then engineer a shift on the data that discards samples based on their age and condition (here Acne). More specifically, we subsample data in order to decrease the bias for younger age in the Acne condition, leading to an increase in age for Acne samples of approximately 10 years. Given that Acne is one of the most prevalent conditions in the dataset, we expect a

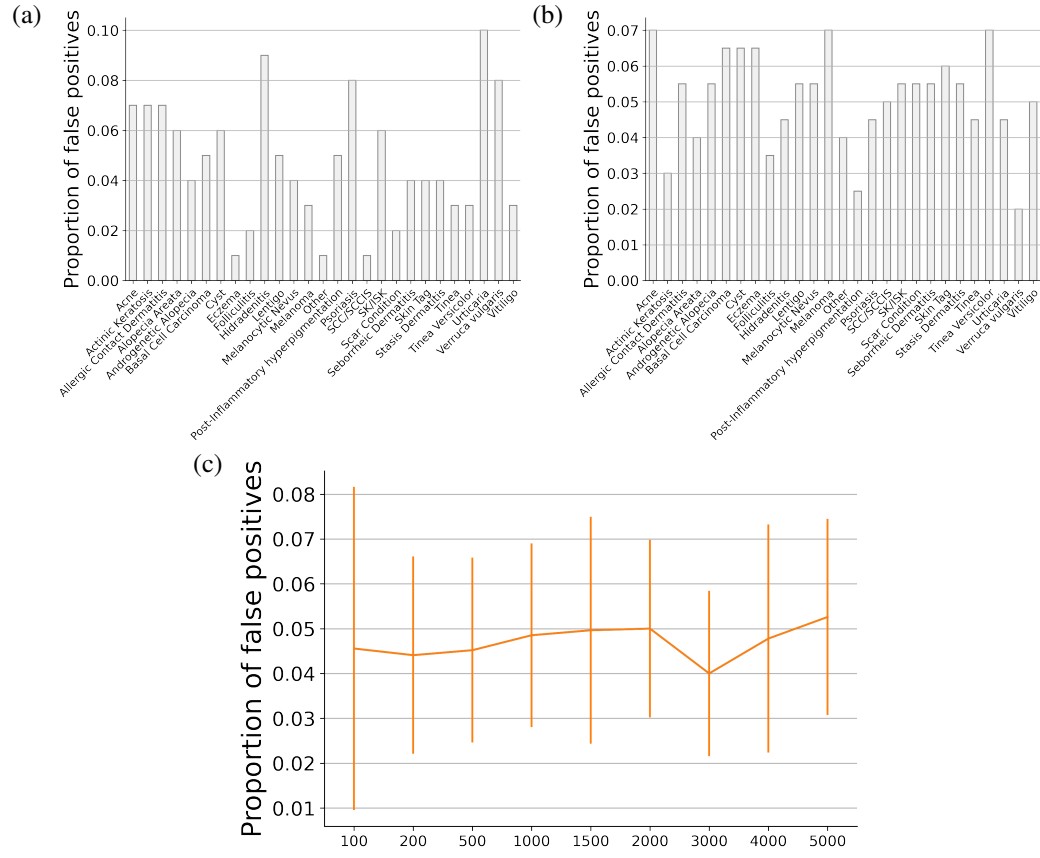

Figure 1: Proportions of false positives when comparing the source data to itself. (a) $P(Y \mid A, S = 0)$ vs $P(Y \mid A, S = 1)$. (b) $P(X \mid Y, A, S = 0)$ vs $P(X \mid Y, A, S = 1)$. (c) $P(Y \mid A, S = 0)$ vs $P(Y \mid A, S = 1)$ when varying the number of samples to estimate $P(S \mid \{\mathbf{V}\})$. In this case, the results are averaged across conditions (with standard deviation).

significant change on age, as well as on $Y$. On the other hand, the images are not modified and hence $P(X \mid Y, A, S = 0)$ and $P(X \mid Y, A, S = 1)$ should not display significant changes.

We observe a significant change in age between $S = 0$ and $S = 1$. There is no significant effect of $S$ on other demographic factors. We do observe a direct effect of $S$ on $Y$, with a strong effect on Acne as expected. On the other hand, no tests display significant differences when comparing $P(X \mid Y, A, S = 0)$ with $P(X \mid Y, A, S = 1)$. This latter result was obtained with both a logistic regression and gradient boosted trees (given the more complex relationship between age and Acne). Therefore, the testing approaches produces a causal graph that mirrors the graph used to engineer the shift, and our procedure is considered faithful.

Finally, we randomly shuffled the labels $y_k$ across the 27 conditions, for each case independently. At the population level, this results in a change in $P(Y)$. As expected, there was no effect of $S$ on $A$. For $Y$, we observed significant differences in $P(Y \mid A, S = 0)$ and $P(Y \mid A, S = 1)$ for 19 conditions out of 27 (Bonferroni corrected). Thanks to the correction, and given that the images were not modified, there are no significant differences between $P(X \mid Y, A, S = 0)$ and $P(X \mid Y, A, S = 1)$.

Overall, our tests are sensitive enough to detect the presence of changes on the affected dimensions, while not displaying an excessive amount of false positives.

## 2  Causal framing of related works

In this section, we review how various transferable fairness approaches that have been proposed in the literature interact with different causal structures of domain shift depicted in Figures 2 and

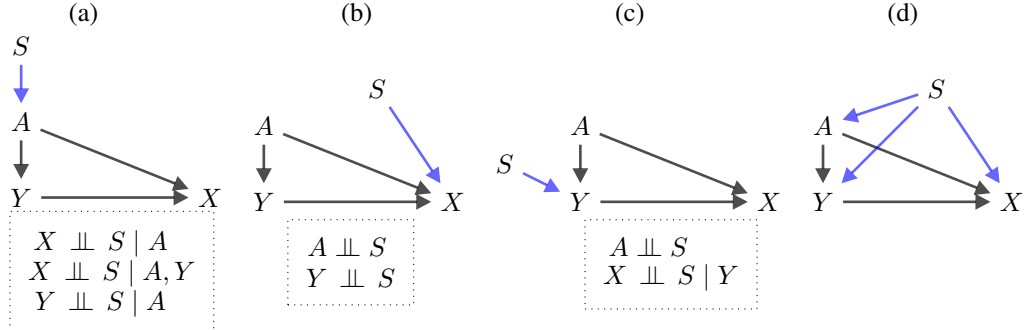

Figure 2: Causal graphs under different distribution shifts when considering an anti-causal prediction task. (a) Exclusive demographic shift where the demographics $A$ are directly affected by $S$. (b) Exclusive covariate shift where the $X$ is directly affected by $S$. (c) Exclusive label shift where $Y$ is directly affected by $S$. (d) Compound shift given by the combination of demographic, covariate, and label shifts.

3. We assume the same sensitive attributes $A$ in the source and target environments, and consider different relationships between the environment and other variables. As in [5, 37], we split the analysis into causal or anti-causal prediction tasks [29]. A prediction task is causal if the features $X$ are causes of the outcome $Y$, and anti-causal if the outcome $Y$ is a cause of the features $X$. Many risk prediction problems are causal, where risk factors may cause an adverse outcome, while many diagnostic problems are anti-causal, where the underlying disease may cause symptoms [5].

A takeaway from this review is that most methods are tailored to particular restricted shift structures, and their useful properties can break down under compound shifts.

## 2.1 Anti-causal prediction tasks

First, we consider anti-causal prediction tasks, in which the outcome $Y$ is a direct cause of the features $X$, and that the sensitive attribute $A$ is a direct cause of $X$ and $Y$ (the reality will likely be more complex and involve other variables in the paths (observed or not)). The assumption that $A$ is a cause of $X$ and $Y$ is a reasonable approximation in healthcare settings: e.g. on average, skin images from men are more hairy than those of women, breasts are visible in chest x-rays, and comorbidities are different across sexes and age ranges. Sensitive attributes can also be related to the label, as prevalence can vary across subgroups (e.g. baldness is more prevalent in older patients) and some conditions might present differently across attributes (e.g. a heart attack in men vs women).

In this context, we consider strategies for learning robustly fair models under exclusive demographic, exclusive covariate, exclusive label and compound distribution shift scenarios represented in the causal graphs of Fig. 2:

**a) Exclusive demographic shift.** In Fig. 2(a), we consider an exclusive demographic shift through the direct path $S \rightarrow A$. In this case, the effects of $S$ on $Y$ and $X$ are intercepted by $A$. Therefore, training a fair model based on $X$ (either via imposing equalized odds [32] or counterfactual invariance with respect to $A$ [37]) leads to a robustly fair model.

**b) Exclusive covariate shift.** In Fig. 2(b), we consider an exclusive covariate shift through the direct path $S \rightarrow X$. In this case, there are multiple causal arrows into $X$ that need to be addressed: $A \rightarrow X$ for fairness, and term $S \rightarrow X$ for robustness to distribution shift. In the case where some labeled target domain data is available, Schumann et al. [30] implement separate independence regularizers to address each path, and demonstrate its effectiveness in settings that match this causal structure. In the fully unlabeled domain adaptation case, the shift is harder to account for. For example, the set of features that satisfy the selection criteria of Singh et al. [32] would only include $A$, leading to a predictor that relies solely on demographics data (Sec. 2.3).

**c) Exclusive label shift.** Lipton et al. [16] propose a technique to detect an exclusive label shift, as well as mitigate it through reweighting, requiring unlabelled target data. This technique can be used in addition to a fairness mitigation technique to lead to a robustly fair model.

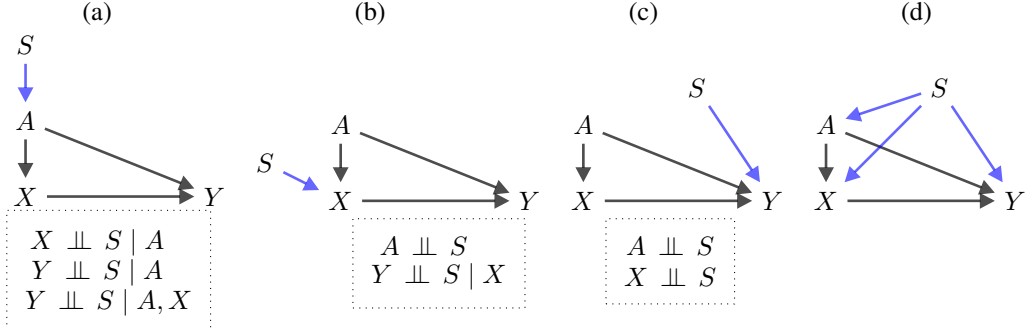

Figure 3: Causal graphs under different distribution shifts when considering a causal prediction task. (a) Exclusive demographic shift, (b) Exclusive covariate Shift, (c) Exclusive label shift, and (d) Compound shift.

**d) Compound shift.** Such a shift leads to multiple factors being affected by $S$, and current techniques would either be ineffective or degenerate to the trivial predictor. While Fig. 2(d) represents the worst case scenario, even simpler combinations of shifts would lead to similar results in practice. For instance, adding a correlation between $S$ and $A$ to Fig. 2(b) leads to a compound shift. In this specific case, mitigation might be possible by taking all intersections of $A$ and $S$ when using [30]. In practice, if the attribute or the shift have multiple discrete values, this will quickly become intractable, especially if distribution matching needs to be applied within each mini-batch. When considering feature selection, [32] would return an empty set for both equalized odds or demographic parity, leading to a trivial predictor. (Partial) mitigation might be obtained through the use of an adaptation technique as proposed in [33, 19].

## 2.2 Causal prediction tasks

We repeat the analysis above for the simplest causal prediction task in which $X$ is a direct cause of $Y$, and assuming that $A$ is a direct cause of $X$ and $Y$. We then consider the following distribution shift scenarios represented in Fig. 3:

**a) Exclusive demographic shift.** As in the anti-causal case, the relationship between $X$ and $Y$ is not affected by $S$ if we impose invariance of $Y$ on $A$. To this end, training a fair model based on the set $\{X, A\}$ would lead to a robustly fair model [32].

**b) Exclusive covariate shift.** In this setting, note that $(Y, A) \perp\!\!\!\perp S \mid X$. Thus, fairness and robustness to distribution shift guarantees can be obtained independently. This setting corresponds to the classic covariate shift scenario treated in much of the domain adaptation literature, where potential solutions are to match the distributions of $X$ across environments by strategies such as reweighting [e.g. 31] or invariant representation learning [e.g. 4]. The advantage of such a techniques is that they only require unlabelled target data.

**c) Exclusive label shift.** While solutions are investigated for anti-causal prediction tasks, label shifts on causal prediction tasks are understudied and the absence of a direct path $S \to Y$ is often an assumption for mitigation techniques [32, 37]. To the best of our knowledge, there is no method proposing formal fairness guarantees under exclusive label shift in causal prediction tasks.

**d) Compound shift.** As for the anticausal case, a compound shift would lead to insufficient or trivial predictors.

## 2.3 Feature selection mitigation strategies

We summarize the results for feature selection, and in particular the method proposed in [32] in Table 1. We observe that exclusive covariate and label shifts cannot be mitigated without excluding the signal, leading to an empty set $\{\mathbf{V}\}$ in some of our simplified cases.

Table 1: Review of the coverage of [32] across different prediction tasks including both $A$ and $S$. We refer to equalized odds for the anti-causal predictive case, and demographic parity for the causal predictive case, as in [37]. Here, $\{\mathbf{V}\}$ is the set of variables to include as inputs to the model.

| Predictive task | Shift | $\{\mathbf{V}\}$ |
|---|---|---|
| Anti-causal | demographic | $\{X\}$ |
| | covariate | $\{A\}$ |
| | label | $\emptyset$ |
| | compound | $\emptyset$ |
| Causal | demographic | $\{X, A\}$ |
| | covariate | $\emptyset$ |
| | label | $\emptyset$ |
| | compound | $\emptyset$ |

# 3   Data and method availability

The dermatology data is not available to the public. The de-identified EHR data is available based on a user agreement at Physionet [10]. The code for extracting Elixhauser and van Walren comorbidity scores from MIMIC-III is available at `https://doi.org/10.5281/zenodo.821872` [12]. We take inspiration from the code made publicly by the authors of [35, 35] and available at `https://github.com/google/ehr-predictions`. We use scikit-learn [22] to estimate the balancing weights.

# 4   Computational resources

Our statistical tests include two main components:

- Estimation of $P(S \mid \{\mathbf{V}\})$: logistic regression or boosted gradient trees are trained 100 times (number of bootstrap samples) for each test. At a maximum, this operation uses 12 Gb of RAM and 2 Gb of memory. All such model trainings are performed in notebooks.

- Summary of $U$: when the dimensionality $l$ of $U$ is larger than a couple of dozens, we summarize $U$ by training a model $f(U) \to Y$. For dermatology, a model training and tuning requires 28 TPU resources over 24-30 hours. For EHR, a model training requires 2 CPU for ~1 hour.

To assess model performance and fairness across multiple distribution shifts, we train 10 replicates of a dermatology model $f(X, A) \to Y$ on the source data, and 10 replicates of an EHR model $f(M, X, T) \to Y$. We also train 10 replicates for the joint training in dermatology.

The training of larger models is performed on an internal cluster.

# 5   Dermatology

## 5.1   Ethics approval and data availability

The images and metadata were de-identified according to Health Insurance Portability and Accountability Act (HIPAA) Safe Harbor prior to transfer to study investigators. The protocol was reviewed by Advarra IRB (Columbia, MD), which determined that it was exempt from further review under 45 CFR 46. The dermatology data is not available to the public.

## 5.2   Model and performance

### 5.2.1   Model architecture

We train 10 replicates from different random seeds of a deep learning model to predict skin conditions from the images $X$, age and sex, with an approach similar to [17, 27]. To train the model, we consider all images pertaining to a case (min 1, max 6). Each image is resized to $448 \times 448$ pixels and encoded with a wide ResNet-101$\times$3 feature extractor initialised using BiT-L pretraining checkpoints [13]. The

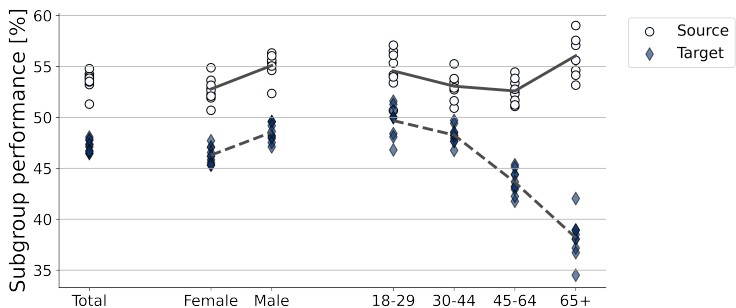

Figure 4: Model performance in dermatology, as estimated via Top-1 accuracy (in %). The plot displays the total performance, as well as performance stratified by sex and by age on the source (circles with plain line) and target (diamonds with dashed line) data. Each marker represents one replicate of the model.

embeddings are then averaged across images and concatenated with the metadata (here age and sex) before passing through a fully connected layer which is followed by classification heads predicting the 26 + 1 conditions [17, 27]. An additional classification head covering a more fine-grained set of 419 conditions, defined on the same examples, is used at train time only.

### 5.2.2 Fairness properties are affected by the environment

Detailed top-3 and top-1 model performance can be found in Tables 2 and 3, respectively. For top-1 accuracy, we observe similar results as for top-3 accuracy: model performance is relatively similar between groups on the source data, but differences become apparent on the target data (Figure 4), especially for age. Interestingly, the target includes 544 paediatric cases, on which the model performs better than in other age groups (top-3: $85.28 \pm 1.32\%$, top-1: $51.53 \pm 1.81\%$).

Table 2: Top-3 model accuracy (in %) in the source and target data, on average across model runs.

| Group | Source | Target |
|---|---|---|
| Total | $88.52 \pm 0.68$ | $79.35 \pm 1.02$ |
| Female | $88.95 \pm 0.93$ (n=1,221) | $79.02 \pm 0.92$ (n=1,115) |
| Male | $87.78 \pm 0.52$ (n=703) | $79.85 \pm 1.28$ (n=728) |
| $[18, 30)$ | $88.51 \pm 1.11$ (n=563) | $81.85 \pm 1.74$ (n=314) |
| $[30, 45)$ | $88.38 \pm 1.02$ (n=548) | $80.21 \pm 0.93$ (n=340) |
| $[45, 65)$ | $88.89 \pm 0.70$ (n=619) | $74.68 \pm 1.45$ (n=419) |
| $[65, 90)$ | $87.84 \pm 1.70$ (n=194) | $68.98 \pm 2.00$ (n=226) |

Table 3: Top-1 model accuracy (in %) in the source and target data, on average across model runs.

| Group | Source | Target |
|---|---|---|
| Total | $53.62 \pm 0.88$ | $47.18 \pm 0.56$ |
| Female | $53.62 \pm 0.88$ | $46.29 \pm 0.77$ |
| Male | $55.09 \pm 1.05$ | $48.54 \pm 0.86$ |
| $[18, 30)$ | $54.49 \pm 1.73$ | $49.68 \pm 1.41$ |
| $[30, 45)$ | $53.12 \pm 1.14$ | $48.26 \pm 0.82$ |
| $[45, 65)$ | $52.68 \pm 0.92$ | $43.63 \pm 1.12$ |
| $[65, 90)$ | $55.52 \pm 2.00$ | $38.19 \pm 1.83$ |

### 5.2.3 Per condition model performance

We further compute model performance per condition when enough data samples are available both in the source and target data. We first observe that the tail of other conditions ('other', Figure 5(a))

shows more similarity with the source, and even improved performance. The fairness patterns for Melanocytic Nevus also seem to be similar across datasets, decreasing with age. On the other hand, SK/ISK shows an increase in performance with age in the target, but not in the source. This last result highlights that the changes in fairness patterns do not only result from changes in condition prevalence. We note that the analyses in this section include relatively low number of samples per subgroup.

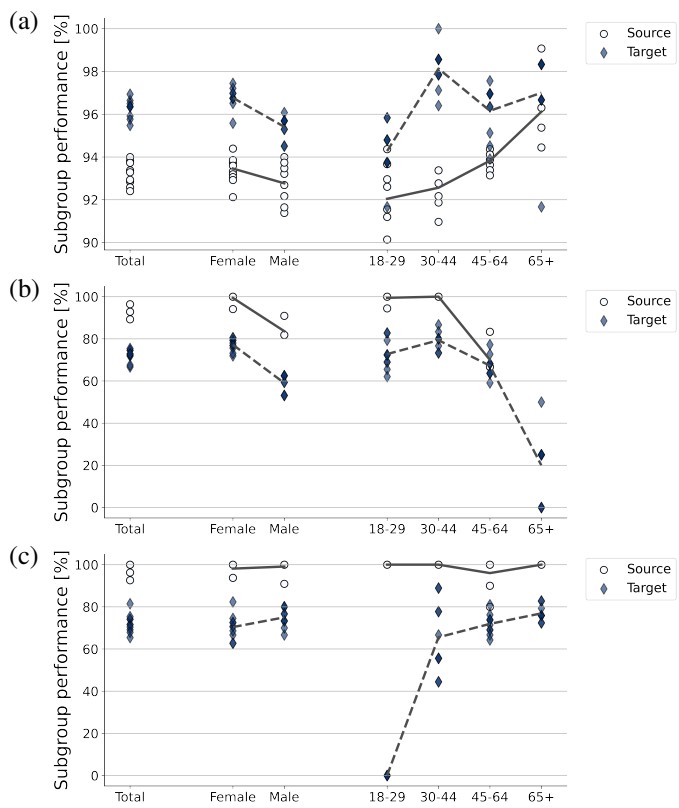

Figure 5: Per condition model performance in dermatology, as estimated by top-3 accuracy (in %). (a) 'Other'. (b) Melanocytic Nevus. (c) SK/ISK.

#### 5.2.4 Fitzpatrick's skin type

In this target dataset, skin type is available as an attribute. We therefore perform similar analyses as performed for age and sex. We first observe that the proportions of cases across the different skin types are different across the source and target datasets (Figure 6(a), t-test: $p < 0.001$).

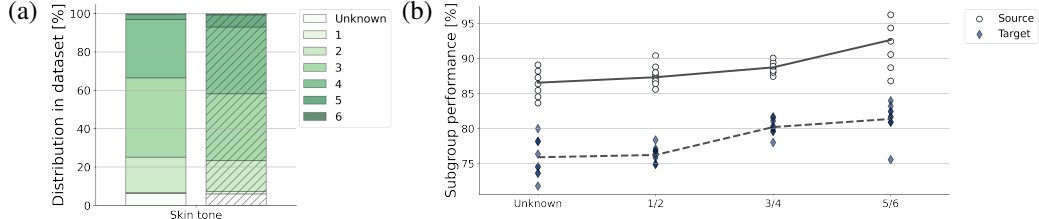

Figure 6: (a) Prevalence of the six Fitzpatrick's skin types in each dataset (left: source, right: target). (b) Top-3 model performance for each subgroup in the source and in the target.

Due to a low number of examples for skin types 1 and 6 (in both datasets), we group skin types according to 'unknown', '1/2', '3/4' and '5/6' before assessing top-3 performance per subgroup. We observe that model performance is relatively similar across the different subgroups, with the

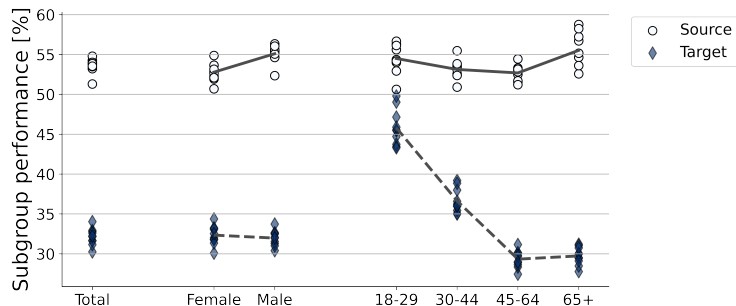

Figure 7: Model performance in dermatology, as estimated via Top-1 accuracy (in %). The plot displays the total performance, as well as performance stratified by sex and by age on the source (circles with plain line) and target (diamonds with dashed line) data. Each marker represents one replicate of the model.

maximum pairwise difference being 6% on the source and 5.5% on the target (Figure 6(b)). We however note that, despite the groupings, the number of examples in group 5/6 remains low (source: $n = 53$, target: $n = 131$). Performance results in this group are hence to take with a grain of salt (as displayed by large variability across seeds), and single conditions cannot be investigated.

## 5.3 Second target dataset: skin cancer clinics in Australia

In this section, we repeat the analyses performed in the main text for another target dataset from skin cancer clinics in Australia and New Zealand that was partially labelled, leading to 21,661 cases used for model evaluation.

### 5.3.1 Fairness properties are affected by the environment

Detailed top-3 and top-1 model performance can be found in Tables 4 and 5, respectively. For top-1 accuracy, we observe similar results as for top-3 accuracy: model performance is relatively similar between groups on the source data, but differences become apparent on the target data (Figure 7).

Table 4: Top-3 model accuracy (in %) in the source and target data, on average across model runs.

| Group | Source | Target |
|---|---|---|
| Total | $88.52 \pm 0.68$ | $70.87 \pm 0.85$ |
| Female | $88.95 \pm 0.93$ (n=1,221) | $72.11 \pm 0.98$ (n=10,195) |
| Male | $87.78 \pm 0.52$ (n=703) | $69.77 \pm 0.80$ (n=11,466) |
| $[18, 30)$ | $88.51 \pm 1.11$ (n=563) | $87.52 \pm 1.15$ (n=1,434) |
| $[30, 45)$ | $88.38 \pm 1.02$ (n=548) | $77.64 \pm 0.79$ (n=4,365) |
| $[45, 65)$ | $88.89 \pm 0.70$ (n=619) | $68.39 \pm 0.97$ (n=8,355) |
| $[65, 90)$ | $87.84 \pm 1.70$ (n=194) | $66.20 \pm 1.44$ (n=7,401) |

Table 5: Top-1 model accuracy (in %) in the source and target data, on average across model runs.

| Group | Source | Target |
|---|---|---|
| Total | $53.62 \pm 0.88$ | $32.14 \pm 0.99$ |
| Female | $53.62 \pm 0.88$ | $32.34 \pm 1.14$ |
| Male | $55.09 \pm 1.05$ | $31.96 \pm 0.91$ |
| $[18, 30)$ | $54.49 \pm 1.73$ | $45.80 \pm 2.14$ |
| $[30, 45)$ | $53.12 \pm 1.14$ | $36.63 \pm 1.45$ |
| $[45, 65)$ | $52.68 \pm 0.92$ | $29.32 \pm 1.01$ |
| $[65, 90)$ | $55.52 \pm 2.00$ | $29.72 \pm 1.05$ |

### 5.3.2 Per condition model performance

We repeat the analyses for the same conditions depicted in Section 5.2.3. We observe that the gaps between groups vary based on the condition, with the long tail of conditions (represented as a single 'other' condition in our task) being seemingly 'fair' on the source, but not on the target (Figure 8(a)). Some conditions like Melanocytic Nevus present disparities across age groups in both datasets (with the caveat of few test samples in the source data), with performance decreasing with increasing age (max gap in source: 24.47%, in target: 39.00%, Figure 8(b)). On the other hand, the SK/ISK condition displays similar performance across age groups in the source, but increasing performance with increasing age in the target (max gap in source: 4.00%, max gap in target: 31.78%, Figure 8(c)). Where the sample size allowed for comparisons, we therefore observe that the gaps across groups are not reproducible between the source and target data.

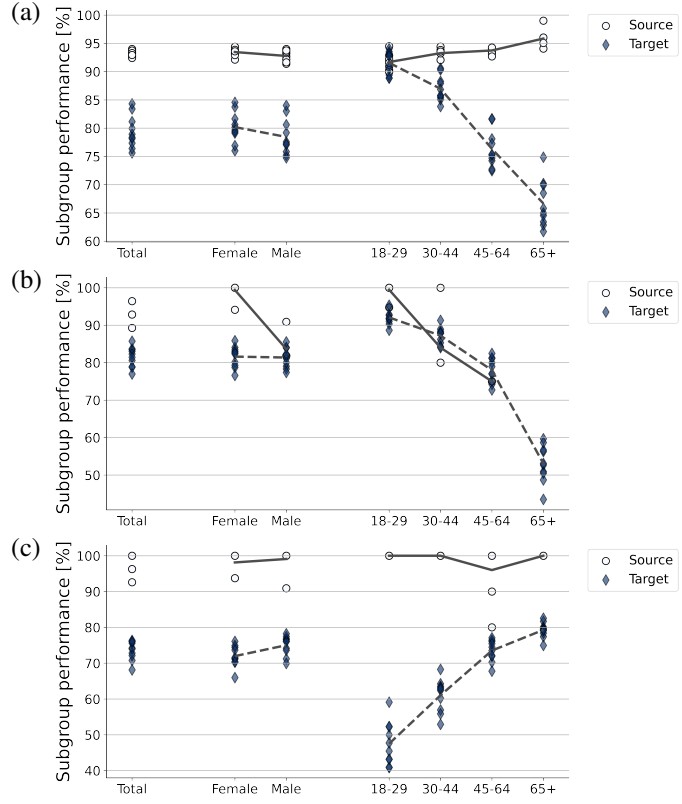

Figure 8: Per condition model performance in dermatology, as estimated by top-3 accuracy (in %). (a) 'Other'. (b) Melanocytic Nevus. (c) SK/ISK.

### 5.3.3 A compound shift

Referring to the same causal graph as in the main text, we apply our testing procedure to this second target set.

Only age and sex[3] are available in both datasets. These features have different distributions across the two datasets (see Fig. 9(b)), with the population in the source data being typically younger (median age: 40 years old, 25% quantile: 27, 75% quantile: 54) than in the target data (median age: 58 years old, 25% quantile: 44, 75% quantile: 70). The source distribution also includes more female patients than the target data (62%, compared to 47%). We therefore see direct effects of $S$ on $A$ (Fig. 9a, in purple).

We then assess whether $S$ directly affects the labels. Based on our testing procedure ($\{\mathbf{V}\} = \{A\}$), we obtain significant differences between $P(Y \mid A, S = 0)$ and $P(Y \mid A, S = 1)$ for 25 conditions

---

[3]Sex is mainly recorded by clinicians, with a small set of the source data containing self-reported sex.

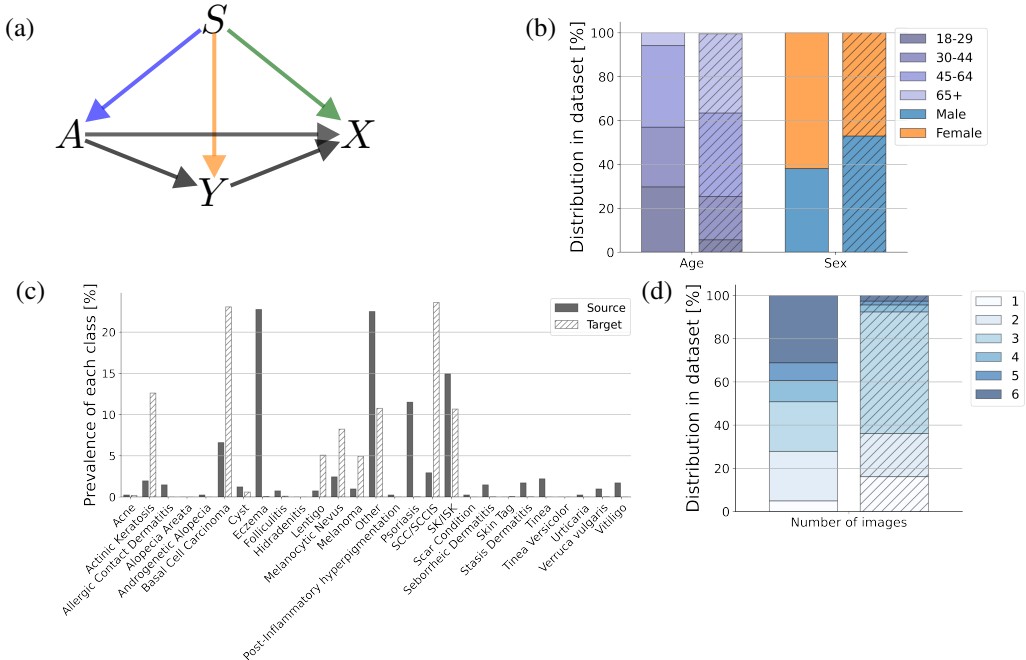

Figure 9: A compound shift in dermatology. (a) Simplified causal structure of the application. $A$ refers to demographic metadata, $Y$ to the skin condition as labelled by clinicians and $X$ represents the set of pictures of the skin pathology. In our setup, the environment $S$ directly affects all variables in the graph. (b) Distribution in the source train and in the target dataset of the sensitive attributes, computed in terms of percentage of pre-defined subgroups. (c) Prevalence of each condition in cases where the patients are female and over 65 years of age. (d) Distribution of the number of images in cases labelled as 'SK/ISK' in older females. The distributions in the source data are represented on the left, and their corresponding distributions in the target data on their immediate right with a hashed pattern.

($p < 0.05$, Bonferroni corrected). Fig.9(c) illustrates the label shift in a specific age and sex subgroup (here females aged 65+, 409 cases in the source, 3,198 in the target data). We see that the source data includes more cases of 'eczema' and 'psoriasis' and that cancerous conditions such as 'basal cell carcinoma' and 'melanoma' are more prevalent in the target data. Our results suggest that the environment also directly affects the labels (orange link in Fig.9(a)). We note that this analysis is limited to the distribution of labels and cannot assess differences in their quality: for instance, the labels for the source data were sourced from multiple clinicians, while some target labels were verified by biopsies. This is an added difference between the two environments (known as annotation shift, [5]) that only in-depth dataset knowledge can bring.

Finally, we consider whether the features of the images themselves (designated by $X$) are directly affected by $S$. Our weighted tests suggest a significant difference between these two distributions ($p < 0.05$ on 21 dimensions, corrected). Fig.9(d) illustrates this difference by computing the number of images per case in the group of older females considered above with cases labelled as 'SK/ISK' ( source median: 3, $n = 61$, target median: 2, $n = 341$). This result suggests the existence of the direct path $S \rightarrow X$ and we add this relationship to the graph (green link in Fig.9(a)).

Based on these different analyses, we conclude that the environment is affecting all the variables in our simplified causal graph.

## 5.4 Joint training

In this section, we test whether joint training across datasets improves the transfer of fairness. We select the setup where the second target dataset (skin cancer clinics in Australia) is available for model development and add it to the source data, while we aim to obtain a robustly fair model on the first target dataset (teledermatology clinics in Colombia). This provides a training set that approximately doubles in size and now includes many more examples of cancer-related conditions.

We first assess whether the distributions of this joint source more closely match the distribution of the target. To this end, we perform the same statistical tests as previously but select half of the source data from each dataset. This simulates an equal proportion of each dataset in the joint distribution.

We observe that $A$, $Y|A$ (5 conditions with $p < 0.05$ corrected) and $X|A, Y$ (2 conditions with $p < 0.05$ corrected) display effects of the environment. Interestingly, the joint source introduces an effect of $S$ on different dimensions than previously. For instance, we now notice a difference in the distribution of cancer-related conditions between the joint source and the target that was not noticeable when comparing the original source (teledermatology clinics in the US) to the target.

In terms of model training, we observe that combining the two datasets for joint training is non-trivial. This is due to the datasets being very different (see section 5.3). We take the simple approach of sampling a batch of each dataset with equal probability. This will oversample the data from the least represented dataset (here the original source). We note that this probability could be optimized over.

We report the results on the test set of the original source, on the test set of the joint source (randomly sampling $1,500$ cases from each dataset), and on the target set (Fig.10). We first observe that the overall performance gap between the source and the target has decreased. However, the fairness gap between source and target remains important, especially for age (max gap between groups: joint source = $4.33\%$, target = $11.4\%$).

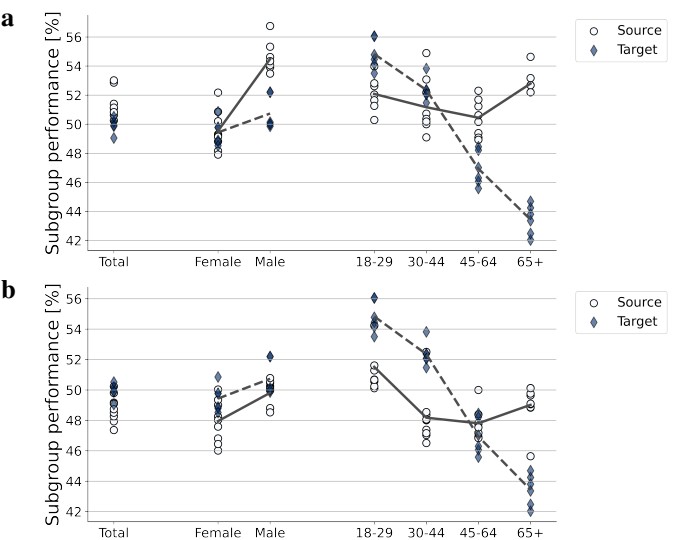

Figure 10: Model performance (top-1 accuracy) after joint training on the source data (circles) and target data (diamonds). (a) Original source test set. (b) Joint source test set.

We therefore conclude that joint training, as it does not introduce invariances to the environment here, does not guarantee an improvement in the transfer of fairness.

# 6 Electronic health records

## 6.1 Dataset

**Data availability:** The de-identified EHR data is available based on a user agreement at Physionet [10].

**Cohort selection:** Our cohort includes patients with age $> 18$ at time of admission and ICU length of stay (LOS) $> 1$ day. We restrict the patient sequences to the first ICU admission, as in the MIMIC-Extract project [38], and additionally restrict to only those admissions with a FIRST_CAREUNIT defined in the MIMIC-III icustays table. This procedure results in a cohort of 28,083 patients.

**Feature and target definition:** The full feature set from [28] is used, leading to 31,303 binary features and 28,048 numeric features. The target is defined as a binary outcome, i.e. remaining ICU LOS $> 3$ days, following the approach used in [38, 23].

Patient demographics are used as metadata for fairness mitigation and evaluation. For illustration, we focus on sex and age (bucketed as in the dermatology application).

**Data splits:** The FIRST_CAREUNIT defined in the MIMIC-III icustays table is used to split the dataset into a source dataset with 17,641 patients (MICU, SICU and TSICU) and target set with 10,442 patients (CCU, CSRU). The source dataset is then randomly split into training (80%), validation (5%), calibration (5%) and test (10%) sets.

## 6.2 Model and performance

As performed in [28, 34, 35], each patient's medical history is converted to a time series of one-hour aggregates including the different structured data elements (medication, labs, vitals, ...) represented by numerical and binary variables. Based on the first 24 hourly aggregates, we predict the binary label for prolonged length of stay using a recurrent network architecture [34, 35]. We then estimate model performance in terms of accuracy across patients and per sex and age subgroups. In addition to per-group metrics, we compute demographic parity and equalized odds.

Table 6 displays the detailed model performance on the source and on the target. On the source data, the model performs to $78.6 \pm 0.7\%$ accuracy, with $\sim 1\%$ performance gap between sex groups and a gap of $7.4\%$ between age subgroups. Performance increases slightly between the two environments, with the model reaching $79.7 \pm 0.9\%$. The gap between sexes increases to $2.7\%$ on the target data while it remains similar for age (7%). In terms of fairness metrics, demographic parity is $0.002 \pm 0.002$ for sex on the source, and $0.016 \pm 0.003$ on the target. It is $0.05 \pm 0.006$ for age on the source, and $0.066 \pm 0.010$ on the target. For both attributes, we hence observe an increase in demographic parity between the source and the target. Equalized odds do not display a significant difference between the two environments, for age or sex.

Table 6: Model accuracy (in %) in the source and target data, on average across model runs.

| Group | Source | Target |
|---|---|---|
| Total | $78.6 \pm 0.7$ | $79.7 \pm 0.9$ |
| Female | $77.8 \pm 0.8$ | $78.0 \pm 0.6$ |
| Male | $79.2 \pm 0.5$ | $80.7 \pm 0.5$ |
| $[18, 30)$ | $84.0 \pm 1.6$ | $82.5 \pm 0.9$ |
| $[30, 45)$ | $82.8 \pm 0.9$ | $84.2 \pm 0.9$ |
| $[45, 65)$ | $78.8 \pm 0.5$ | $83.2 \pm 0.6$ |
| $[65, 90)$ | $76.6 \pm 0.7$ | $77.2 \pm 0.6$ |

## 6.3 A compound shift

### 6.3.1 The ICU unit $S$ affects the comorbidities $M$

We define comorbidities according to [9, 36] using code in [12], obtaining a set of 30 comorbidities associated with each patient (multi-label). In this work, we refer to comorbidities as a 'summary' of ICD codes that represent the patient's medical history prior to the current admission.

**Statistical testing**: We define IPW weights based on a logistic regression that predicts the environment based on 5,000 admissions from the source and 5,000 admissions from the target data. The classifier performs with an accuracy of 58.6% on a left-out test set comprising 20% of the data. For each comorbidity, we assess whether its prevalence is significantly affected by the environment using weighted tests. We observe that 5 comorbidities lead to significant results (after Bonferroni correction for multiple comparisons): pulmonary circulation, peripheral vascular, liver disease, metastatic cancer and solid tumor. We however caveat this analysis by the low number of patients whom have recorded comorbidities prior to admission, leading to many tests being invalid.

### 6.3.2 The ICU unit $S$ affects the treatments $T$

**Definition of treatments**: We identify treatments based on the 'MedicationRequest' field in the FHIR representation [25] of MIMIC. For each of those treatments, we construct a summary by

counting the number of hourly requests in the first 24 hours after admission. This summary is used for illustrating specific medications (beta-blockers, vasopressors/inotropes and ACE inhibitors) and to control for treatments in section 6.3.3. Given that this representation includes 5,873 dimensions, we build a further 1-dimensional summary [16] by training a model that predicts LOS from the time series of the medication requests on the source training split. This model reaches an accuracy of 76.9% (AUPRC: 0.54, AUROC: 0.75) on the source test split.

**Statistical testing**: We assess whether treatments are directly affected by $S$ by computing weights to balance $\{A, M\}$. We note that this analysis is limited as comorbities are not reported for all patients. Nevertheless, a binary model discriminating between environments based on $\{A, M\}$ leads to 62% accuracy. Given the dimensionality of treatments, we perform a weighted test on the summary of treatments. Considering all medication, we observe that patients receive different treatments in different units (weighted test, $p = 0.015$), hence $T \not\perp\!\!\!\perp S | \{A, M\}$.

As an illustration, we estimate what proportion of patients receives treatments such as beta-blockers, vasopressors/inotropes and/or ACE inhibitors (Figure 11) for patients with (a) vascular comorbidities, and (b) tumor history.

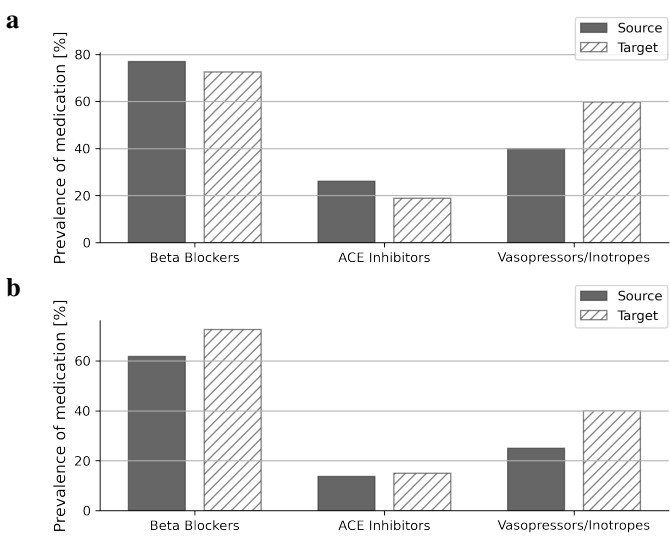

Figure 11: Per comorbity prevalence of treatment (in %). (a) Peripheral vascular comorbidities. (b) Solid tumor.

### 6.3.3  The ICU unit $S$ does not affect the length of stay $Y$

We refer to the summary of treatments of dimension 5,873 built above to define $T$ and build the feature set $\{A, M, T\}$ to estimate overlap weights such that we can test whether $P(Y|A, M, T, S = 0) = P(Y|A, M, T, S = 1)$. The treatment features with non-null values are normalized to be in the range $[0, 1]$ before we build a classifier that distinguishes between the two environments. The model achieves 87.2% accuracy in predicting the environment. We derive the weights for each data point in the two environments and assess with a weighted t-test whether the prevalence in the two environments is similar. Our results show that the ICU unit might not affect the prevalence of the length of stay ($p = 0.18$). We however note that our analysis is limited by the large dimensionality of $\{A, M, T\}$ to build the weights. Indeed, the gradient boosted tree model $P(S|A, M, T)$ can then rely on many different signals to build its predictions [7] and display high variance across bootstraps. This would result in an under-powered test.

### 6.3.4  The ICU unit $S$ does not affect the labs and vitals $X$

Based on the causal graph, we use the same IPW weights as in Section 6.3.3. We build a summary of $X$ by predicting $Y$ from all features that are not comorbidities, demographics or treatments. This recurrent model reaches 77.84% accuracy (AUPRC: 0.62, AUROC: 0.80) on the source test split. Based on this summary, we observe that labs and vitals do not display evidence of being affected by the environment ($p = 0.45$).