# OpenReview forum: "Diagnosing failures of fairness transfer across distribution shift in real-world medical settings"
_NeurIPS.cc/2022/Conference — NeurIPS 2022 Accept_

### Official Review · Reviewer_Ezbv · 2022-07-09

**Rating:** 7
**Confidence:** 3
**Soundness:** 3 good
**Presentation:** 3 good
**Contribution:** 3 good

**Summary:**

The authors deal with the problem of diagnosing fairness transfer under distribution shifts. For this they use a causal framework and statistical hypothesis testing to identify the type of distribution shift. Their work relies on having the labels of sample of out-of-distribution data to correctly identify the type of shift.

**Questions:**

A synthetic experiment can be switching the labels of the test set. In this way one can have a pure concept shift, and if done only with protected attribute subgroups it can affect fairness. Would the method propose on this paper diagnose correctly this type of shift?

A strategy used in order to post-process the prediction to enforce fairness metric is Alabdulmohsin[4], the authors observe that fairness mitigating on source data leads to no improvement or worsening on test data. I wonder if simpler methods such as [https://www.nature.com/articles/s42256-021-00396-x] have the same results.

**Limitations:**

A common underpinning of much work in ML fairness is that the metric selection is not justified from a socio-technichal perspective. In this work the authors acknowledge that the chosen metrics do not consider some social factors.

The authors also acknowledge that the solution proposed depends on the number/quality of samples and that it should be perceived as to diagnose fairness transfer failures.

**Strengths And Weaknesses:**

The paper deals with a very relevant topic.

The formalization of the problem is quite clear, simple, and suits very well the task at hand.  The paper is well written and easy to understand. Even if it has a long appendix, the main body is self-contained and does not rely on the appendix.

The paper structure is not the typical one, with a related work section on page 7 and a discussion section that spans for over a page.
In the discussion section, there is a set of remedies. I am unsure if a set of remedies can be included in a scientific paper.


Line 101 -- do Rabanser [53] use bootstrapping techniques to do dimensionality reduction?

---

> ### Author Response · Authors · 2022-08-01
> **Point by point response**
>
> We thank the reviewer for their positive comments and for their suggestions. We respond below, with the changes made to the text.
>
> - **Line 101 -- do Rabanser [53] use bootstrapping techniques to do dimensionality reduction?**
>
> The bootstrapping only affects the statistical testing, not the dimensionality reduction. This means that dimensionality reduction is performed independently, and that the summaries are then fed into the statistical testing that involves bootstrapping. We have amended the text to make this clearer:
>
> Caption of Algorithm 1:
> *“Algorithm 1: (Conditional) independence testing assessing the nature of shift S on a single variable U ∈ G. U represents the feature values or its summary if high-dimensional.”*
>
> Questions
> --------------
>
> - **A synthetic experiment can be switching the labels of the test set. In this way one can have a pure concept shift, and if done only with protected attribute subgroups it can affect fairness. Would the method proposed in this paper diagnose this type of shift?**
>
> Thank you for suggesting this experiment. Using the dermatology source data, we have randomly shuffled the soft label across the 27 conditions, for each case. This leads to a change in the prevalence of the conditions, not conditioned on $A$. While there are no changes in P(A) due to S (by design), we observe a significant difference between $P(Y | A, S=0)$ and $P(Y | A, S=1)$ for 19 conditions out of 27 (Bonferroni corrected). Thanks to the correction, and given that the images were not modified, there are no significant differences between $P(X | Y, A, S=0)$ and $P(X | Y, A, S=1)$.
>
> We have added this result in the Appendix:
> *“Finally, we randomly shuffled the labels $y_k$ across the 27 conditions, for each case independently. At the population level, this results in a change in $P(Y)$. As expected, there was no effect of $S$ on $A$. For $Y$, we observed significant differences in $P(Y \mid A, S=0)$ and $P(Y \mid A, S=1)$ for 19 conditions out of 27 (Bonferroni corrected). Thanks to the correction, and given that the images were not modified, there are no significant differences between $P(X | Y, A, S=0)$ and $P(X | Y, A, S=1)$.”*
>
> - **A strategy used in order to post-process the prediction to enforce fairness metric is Alabdulmohsin[4], the authors observe that fairness mitigating on source data leads to no improvement or worsening on test data. I wonder if simpler methods such as [https://www.nature.com/articles/s42256-021-00396-x] have the same results.**
>
> Thank you for this comment. In this work, we have selected a method that comes with a technical justification and guarantees a solution that enforces the desired fairness criterion (tested with demographic parity and equalized odds for EHR). It is possible that different fairness mitigation strategies lead to different fairness transfers (line 336). However, if a strategy does not amend the BCN in a way that guarantees fairness transfer (as specified in our analysis in Appendix A.4), the specific method used for fairness mitigation would not change that result (line 337).

---

### Official Review · Reviewer_aaBt · 2022-07-11

**Rating:** 7
**Confidence:** 4
**Soundness:** 3 good
**Presentation:** 4 excellent
**Contribution:** 3 good

**Summary:**

The paper proposes an approach to detect distribution shifts between two environments or settings that could lead to the failure of transfer of fairness properties across the settings. The approach adopts the joint causal framework to identify the direct effects of the environment variable on other variables at hand. The paper also suggests some mitigation strategies related to post-deployment, data collection, and outcome definition in case distribution shifts make it challenging to transfer fairness properties. Experiments on two real-world health tasks: 1) predicting skin conditions in dermatology and 2) clinical risk prediction from EHR highlight the merits of the proposed approach.

**Questions:**

1. [1] also discuss detecting shifts across environments using a causal discovery approach (FCI) that also adopts conditional independence tests. This is another line of work closely related to this paper and would be helpful to situate the current approach with regards to this work/discuss the differences.

2. In the dermatology task, what is the dimension of X considered? How would the dimensionality affect the results?

3. In figure 2d, does the X-axis represent the plots for the source and target? It's a little unclear as to what the number of images signifies here.

4. Reference missing on line 174.

5. Check if the references for path-specific effects (line 329) are appropriate. [2] is related to this.



References
[1] Singh, Harvineet, Vishwali Mhasawade, and Rumi Chunara. "Generalizability challenges of mortality risk prediction models: A retrospective analysis on a multi-center database." PLOS Digital Health 1.4 (2022): e0000023.
[2] Mhasawade, Vishwali, and Rumi Chunara. "Causal multi-level fairness." Proceedings of the 2021 AAAI/ACM Conference on AI, Ethics, and Society. 2021.


**Limitations:**

The authors have discussed some limitations of the work.

Some other concern is the lack of complete data, especially when considering multiple environments/settings in healthcare such as social determinants of health. This is pertinent since social determinants are known to shift across environments and would thus lead to failure of fairness transfer.

One more concern is when considering multiple sources or multiple target environments. As the number of environments increases the chances of shift would also increase.  While the robust training could be done to incorporate multiple source environments,  assessing the performance across multiple target environments is crucial to understand under what kind of settings is the model fair.

**Strengths And Weaknesses:**

Strengths:
1. The paper is generally well written with a clear description of the proposed approach and the empirical details.
2. The work touches on an important aspect of the transfer of fairness notions, especially in healthcare where there is not enough clarity on which of the fairness notions are more suitable. The topic is of importance to the community and the said approach is one such mitigation/investigative tool. There is potential for further advancement in this area that is touched upon here and it is thus significant.

Weakness:
1. One potential weakness as illustrated is the dependence on sample size which is known to affect conditional independence testing.
2. There are some parallel approaches that touch upon identifying how shifts relate to model performance in the healthcare setting; it would be useful to discuss the current approach with respect to such methods (more on the approaches below).

---

> ### Author Response · Authors · 2022-08-01
> **Point by point response**
>
> We thank the reviewer for their positive assessment of our work and their interesting comments. We respond to your suggestions and questions below.
>
> Questions
> --------------
>
> - **"[1] also discuss detecting shifts across environments using a causal discovery approach (FCI) that also adopts conditional independence tests."**
>
> Thank you for pointing us to the recent work in [1]. This is indeed a very relevant reference that we have now included. The major differences between our approach and [1] are that (a) FCI and MMD can be applied to low numbers of dimensions, while we deal with high-dimensionality data, (b) the MMD tests are done on marginal distributions, (c) we provide a causal analysis of failures of fairness transfer with respect to prior mitigation work, which can guide the selection of potential mitigation strategies.
>
> We have included the citation and added the following to section 5:
> *“Recently, Singh et al. (2022) investigate differences in source and target distributions using squared maximum mean discrepancy (Gretton et al., 2012), but do not take the causal structure into account. To differentiate direct from indirect effects of $S$ on variables $U$, they perform causal inference and visually inspect the obtained graph. This method can only be applied to low-dimensionality datasets.”*
>
> - **"In the dermatology task, what is the dimension of X considered? How would the dimensionality affect the results?"**
>
> The dermatology images are resized to 448 × 448 pixels. We have added this information in the Supplement.
>
> In both datasets, the dimensionality of X is large. There are two potential effects of large dimensionality: (1) to build a reliable summary of these features to perform the test on, and (2) to use such features to derive the weights based on P(S|X) if needed. For (1), Rabanser et al. investigate different summary strategies. As we mention on line 696, our approach does not impose this choice. The goal is therefore to obtain a reliable summary. In both the EHR and dermatology work, we refer to X → Y models and obtain models that perform well on a separate source test set (see Supplement). However, the summary could become “dominated” by specific dimensions. For (2),  we have found that using more features led to higher performance in discriminating between environments (e.g. for EHR: using {A,M}, the performance is 76.9% while using {A,M,T} leads to 87%). However, model performance cannot assess whether the correct / all relationships between features and environments are captured. Therefore, if the variance is high, the test might be underpowered across bootstraps. This is a possible explanation for the EHR tests on X and Y conditioned on >5,000 treatments.
>
> We have detailed our statement in the Supplement:
> *“We however note that our analysis is limited by the large dimensionality of $\{A, M, T\}$ to build the weights. Indeed, the gradient boosted tree model $P(S | A, M, T)$ can then rely on many different signals to build its predictions (D’Amour et al., 2020) and display high variance across bootstraps. This would result in an under-powered test.”*
>
> - **"In figure 2d, does the X-axis represent the plots for the source and target?"**
>
> Apologies for this confusion: every case can have minimum 1 to maximum 6 images (variable). All images for a case are embedded and their average is taken. Hence, the number of images per case likely affects the signal-to-noise ratio. This plot shows that on average, this number is different across datasets. We have modified the legend to make this clearer:
> *“(d) Distribution of the number of images in cases labeled as `SK/ISK' in older females. Each case includes min. 1, max. 6 images whose embeddings are then averaged.”*
>
> - **Reference missing on line 174**
>
> Thank you. This reference is a website. For clarity we have used a footnote instead (noa means no authors).
>
> - **Check if the references for path-specific effects (line 329) are appropriate. [2] is related to this.**
>
> Thank you for this. There seemed to be a compilation mishap as we did mean to cite Chiappa and [2].

---

> > ### Author Response · Authors · 2022-08-01
> > **part II**
> >
> > Limitations
> > -----------------
> >
> > - **Some other concern is the lack of complete data**
> >
> > Thank you for this comment. We touched upon this phenomenon at line 322 regarding missing unobserved factors across environments, and at line 324 around missingness patterns. We have added the following to line 322:
> > *“We are also limited by which factors are observed across all environments, and cannot assess the changes in unobserved variables (e.g. social determinants of health).”*
> >
> > - **One more concern is when considering multiple sources or multiple target environments.**
> >
> > Thank you for this comment. Indeed, our approach refers to a “source” domain and one target domain. While we can perform pairwise comparisons between the source and every possible target, extending the method to a general form of S would be desirable. This could be considered by using permutation weights as described in [Arbour et al.]. We have added this limitation as “future work”:
> >
> > *“In addition, extending the testing strategy to compare multiple (discrete or continuous) versions of the environment $S$ would be valuable.”*

---

### Official Review · Reviewer_4fVY · 2022-07-11

**Rating:** 6
**Confidence:** 3
**Soundness:** 3 good
**Presentation:** 3 good
**Contribution:** 2 fair

**Summary:**

This paper establishes a causal framing to characterize the effects of distribution shift on a trained model's fairness. A statistical test building from various factors of conditional independence, enabled through the assumed causal structure, is used to understand what factors of a dataset are unduly affected by the distribution shift. This knowledge illuminates the possible reasons a model's fairness qualities are not maintained after transfer which in turn motivate possible mitigation strategies. The proposed statistical test is then used to analyze model performance on two separate real-world datasets.

# UPDATE: After discussion period

I'm grateful to the authors for their patience to address my misunderstandings and willingness to improve the writing of the paper. After re-assessing the paper, I am happy to improve my score and recommend acceptance of this paper.

**Questions:**

Unfortunately, I am not sure whether the authors will be able to sufficiently address the major concerns that I raised in my review above that would allow me to raise my score without re-submission. There are core issues to the framing and narrative of the paper necessitating a full revision as well as some additional experimental analysis that would need to be re-reviewed before acceptance for publication. This process is not supported by the reviewing timeline for NeurIPS.

I do have some questions though that may help me understand whether I should reconsider my assessment and if I misunderstood the paper in any way.

In the authors view, what are the major intended contributions of this paper (see first major points in the section above)?

How does this paper differentiate from Creager, et al (2020 ICML)? Is it only in the use of real-world medical datasets and the corresponding simple prediction tasks?

How sensitive is Algorithm 1 to the quality/accuracy of the models used internal to the statistical test?

What efforts were taken in the construction of the experimental subpopulations to create a fair comparison of model performance?

**Limitations:**

The major limitations of this paper have been listed by the authors in the final paragraphs of the paper. They have clearly outlined major challenges from a data perspective that limit the feasibility of mitigation strategies as well as overall performance assurances with regards to transfer when facing distribution shift.

However, it is unclear what kinds of assumptions and limitations that exist for the development and execution of the statistical tests in Alg. 1. I can guess that the data needs to admit some form of conditional independence estimation (which might be a fairly strong assumption on its own) as well as be able to adequately estimate the dataset source (predicting $P(S|pa(U))$).

**Strengths And Weaknesses:**

It's clear that a lot of work has been done in pursuing the research underlying this paper. There are thorough empirical investigations as well as substantial scholarship to properly anchor the work. The statistical test, built around predicting the binary indicator source of the dataset, appears to be a helpful tool. It unfortunately isn't given enough space to be sufficiently introduced or analyzed. I will try, to the best of my ability, to outline where I feel that the paper is strong and provide my view on its weaknesses with some recommendations for improvement along the lines recommended in this form ("originality, quality, clarity, and significance"). All recommended literature beyond those referenced in this paper will be included at the bottom of this section.

**To be up front, it is unclear what the intended contributions of this work are meant to be.** The causal framing of datasets and the effects of environment (e.g. source and target dataset) has been established previously (see Adragna, et al '20; Madras, et al 2019; Creager, et al '20, '21, _some of these citations were missing surprisingly_). The statistical test has some merit as a potential contribution but is not presented convincingly as such, nor is it used to any extensive effect in this paper. Finally, the mitigation approaches outlined in the Appendix are all established in the literature and are discussed, _even in this paper_, to not be realistic or particularly helpful. In this regard, it is unclear how to assess the paper's originality. Perhaps the extension of this lines of previous inquiry to real-world medical datasets is the major contribution? This is a noteworthy effort that is to be commended but the tenor of the findings only serve to reinforce well understood complexities with medical data (e.g. the type of distribution shift is compound and possibly too complex to adequately or feasibly mitigate).

Another issue along the lines of originality here is that the ML-pipeline remedies (lines 293-315) to address fairness challenges (among other issues raised by distribution shift) are almost entirely pulled from Chen, et al (2021) without proper attribution. At first read this section seemed like an odd departure from the narrative and form of the preceding sections of the paper. After re-reading the paper a second time, it comes across as a mea culpa trying to mask technical deficiencies in the work by redirecting the focus away from the results.

**The framing of "fairness transfer" is odd**. While the fairness of trained models is an important characteristic to monitor and assess, it strikes me as an odd framing to place model performance in as focused on "problems" with the data. The issue at hand are artifacts of the model training and its generalization capacity. Issues of fairness, while being spurred by characteristics of the data (especially under transfer), largely arise through the model overfitting to "degenerate" or local modes in the global data distribution. Placing this entire blame on the data doesn't seem appropriate. Perhaps, it might be better to frame this challenge as ways the datasets are insufficient to ensure equitable model performance across distribution shift? I acknowledge that the term "fairness transfer" is present in the literature, of which we can't control. But, there is some improvement possible to the way this literature frames and addresses these problems. I really enjoyed the attempts this paper makes to 1) identify the challenges to maintaining model fairness that arise due to distribution shift (re-affirming Nestor, et al 2019), 2) evaluate the effects of distribution shift on cross-sectional model performance, 3) propose possible mitigation strategies and 4) recommend possible mediation strategies to the ML pipeline when facing these types of challenges. I just wish that the narrative and structure used in this paper made the actual challenges more apparent without resorting to shorthand that isn't wholly indicative.

**The "diagnosis" of the distribution shifts in the real-world datasets is underwhelming.** At worst, the analysis described in the paper can be viewed as elementary data exploration. It was surprising that after introducing Algorithm 1, that it wouldn't be used as readily for testing the independence of each variable in the dataset. Simple distributional analyses of the data and its subgroups seem to dominate the assertion (and pretty well understood fact a priori) that the type of shift is compound. The mention of the statistical tests is a footnote and seems to be largely kept quite vague. I suppose that I was hoping for a bit more tangible evidence whether through tables or charts about how the statistical tests were used and how they were assessed to be meaningful or not (especially since they rely on prediction models internally, how accurate or useful were those by the way?!). Parenthetically stating that something was significant or not based on the p-value doesn't necessarily provide enough information about the validity of the underlying statistical test. It largely feels that the analysis of distribution shifts could be does by estimating the simple disease prevalence among subgroups or otherwise measuring the homogeneity of the dataset using something like optimal transport dataset distance (OTDD, Alvarez-Melis and Fusi, 2021).

**The procedure and execution of the proposed statistical test is unclear.**
As far as I understand the statistical test proposed in Algorithm 1 is built around learning predictive models of the data source, conditioned on the causal parents of the dataset. The procedures in setting up and evaluating these models is not made clear, nor are the underlying effects of possibly incorrect models on the overall ability of the statistical test to adequately reject the null hypothesis. As the core technical contribution (as far as I can tell), it is surprising that the mechanisms and analysis of this algorithm are not featured in any substantive manner. Additionally, it's unclear whether Algorithm 1 is used on each variable in the BCN individually or all at once.

Beyond this, it is difficult to adequately gauge the validity or helpfulness of the proposed statistical tool with the datasets used in the paper. Here, the design of synthetic examples where a subset of the variables are affected by the shift (rather than *all* of them) would be very insightful. It would be helpful to see use cases where the proposed test does work well and as designed/intended. Right now, going from idea to implementation in complex settings, the presumed impact of the approach is significantly lacking.

Additional points where the clarity of the paper could be improved:
- The use of notation in Algorithm 1 is terse and not introduced anywhere. It's difficult to follow and understand. This is made harder by compounding the use of 'S' to be both the sample from the datasets and the source/target indicator.
- It would be instructive to outline the BCN basic structure or concepts using a figure early on. This could be particularly useful in the design and execution of synthetic examples where the causal graph is modified accordingly.
- It's unclear what the training/testing procedure is to assess model fairness before and after the dataset shift. Fairness properties of a model depend significantly on the training mechanism of the model. The absence of any discussion along these lines is concerning.
- In the EHR experiments, only 5 comorbidities were accounted for in the target population. How were these chosen? Why only those 5?


**Re: EHR experiments** It is stated that many hospitals do not have specialized ICU departments. This assertion is not supported by citation and seems speculative and misleading. This is made more apparent when the experiment utilizes MIMIC data that is derived from a hospital that *does* have specialized ICU departments. The poor model performance between the two subsets of data is not at all surprising since the subsets of the data cover very disparate patient populations. The design of these experiments is not very intellectually honest without proper description or justification. It can be understood if the subpopulations were designed to be an apparent and extreme example of distribution shift but without stating this, it appears that the authors are trying to construct a dramatic example to garner more significance. The conclusions made in lines 216-221 are exceedingly obvious. Of course the type of ICU a patient is admitted to affects the underlying performance of a model trained on a separate population. There are basic factors due to underlying health conditions, whether the procedure was elective or urgent, hosptial procedures, etc. that confound the presentation of data for use in predictive settings. By transferring between subpopulations designed to be as far apart as possible along some of these confounding dimensions (especially when these things are well understood within MIMIC--CSICU vs. MICU for example), there are some significant concerns about the intentions of the authors in using these experiments. Again, if this was clearly set out and justified from the outset it would be more more acceptable (yet, still inappropriate) that these empirical claims were made in this paper.

#### _**Recommended Literature**_

Madras, D., Creager, E., Pitassi, T., & Zemel, R. (2019, January). Fairness through causal awareness: Learning causal latent-variable models for biased data. In Proceedings of the conference on fairness, accountability, and transparency (pp. 349-358).

Creager, E., Madras, D., Pitassi, T., & Zemel, R. (2020, November). Causal modeling for fairness in dynamical systems. In International Conference on Machine Learning (pp. 2185-2195). PMLR.

Alvarez-Melis, D., & Fusi, N. (2020). Geometric dataset distances via optimal transport. Advances in Neural Information Processing Systems, 33, 21428-21439.

---

> ### Author Response · Authors · 2022-08-01
> **Point by point response - Part I**
>
> We thank the reviewer for their comments. We hope that our point by point response clarifies our intent and results.
>
> Contributions
> -------------------
> - **“The causal framing of datasets and the effects of environment (e.g. source and target dataset) has been established previously “ [...] “Perhaps the extension of these lines of previous inquiry to real-world medical datasets is the major contribution? This is a noteworthy effort that is to be commended but the tenor of the findings only serve to reinforce well understood complexities with medical data (e.g. the type of distribution shift is compound and possibly too complex to adequately or feasibly mitigate).”**
>
> We do not claim to be the first to point issues with fairness transfer. Line 16 clearly mentions this, citing example papers in the field. However the implications on the field of medical AI are not well understood by this community. This is further illustrated by the recent publication of Singh et al. (April, 2022) pointed out by reviewer aaBt. Therefore, showing these results on medical applications is important for the field of medical AI.
>
> Regarding references: we aim to illustrate our points with example citations in the field from diverse sources and do not claim to be exhaustive. In the references pointed out by the reviewer, Madras et al., 2019 addresses the issue of fairness through causal modeling, but does not address its intersection with distribution shift. Creager et al., 2020 discusses fair policy learning in dynamical systems, which is closer to the topic of robustly fair models. We have added Creager et al., 2020, and discuss below its differences with our work.
>
> - **“The statistical test has some merit as a potential contribution but is not presented convincingly as such, nor is it used to any extensive effect in this paper.”**
>
> The tests are not a side point but the main contribution of our work. Their description, usage and implications are made explicit throughout our manuscript (lines 28, 89, 132-167, 192-221, section 5). We report the full methods and validation in Appendix given space constraints. Most of our results, analysis of related work and discussion reflect the impact these tests can have.
>
> - **“Another issue along the lines of originality here is that the ML-pipeline remedies (lines 293-315) to address fairness challenges (among other issues raised by distribution shift) are almost entirely pulled from Chen, et al (2021) without proper attribution. At first read this section seemed like an odd departure from the narrative and form of the preceding sections of the paper. After re-reading the paper a second time, it comes across as a mea culpa trying to mask technical deficiencies in the work by redirecting the focus away from the results.”**
>
> We strongly disagree with the reviewer’s interpretation of this section. First, we cite Chen et al. and did not “pull” conclusions from their paper. Second, points 2-5 refer directly to the issue of fairness transfer, reframing suggestions in Chen et al under this focus, but also including references to our tests. This section is not a mea culpa for deficiencies in our work, but rather aims to broaden the discussion outside of purely technical approaches to fairness transfer issues. This is actually going beyond what most technical works discuss, bringing more ethical considerations along the ML pipeline. We believe this is an important addition to the discussion.
>
> To make attribution clearer, we have modified our text:
> “Our work highlights real-world challenges that prevent the development of robustly fair models. Given that each task leads to different complexities in terms of causal graph, expected shifts, and fairness requirements, these findings support looking beyond purely algorithmic solutions. *Considering the whole ML pipeline, we can re-interpret the remedies discussed in Chen et al. (2021) for fairness transfer: [...]* ”
>
> The framing of "fairness transfer" is odd
> ------------------------------------------------------
> - **“While the fairness of trained models is an important characteristic to monitor and assess, it strikes me as an odd framing to place model performance as focused on "problems" with the data.”**
>
> We are not claiming this is a data issue, rather that one or multiple datasets used as source cannot capture the whole data generation process and are hence susceptible to shifts. In the same way, one cannot claim that issues arise from models only. It would be helpful if the reviewer could point out which parts of the text refer to fairness transfer as a data issue, and/or make suggestions for improvement as it is unclear what modifications are requested.

---

> > ### Author Response · Authors · 2022-08-01
> > **Point by point response - part II**
> >
> > - **“ I really enjoyed the attempts this paper makes to 1) identify the challenges to maintaining model fairness that arise due to distribution shift (re-affirming Nestor, et al 2019)”**
> >
> > Nestor et al., 2019 highlight issues of robustness with EHR models, not of fairness transfer. Appendix F describes subpopulation stratification results but does not assess the fairness of the models before and after the EHR system change, and does not guarantee the model fairness before the change.
> >
> > The "diagnosis" of the distribution shifts in the real-world datasets is underwhelming
> > ----------------------------------------------------------------------------------------------------------------
> >
> > - **“It was surprising that after introducing Algorithm 1, that it wouldn't be used as readily for testing the independence of each variable in the dataset.”**
> >
> > We have used algorithm 1 for every statistical comparison, apart from the marginal comparisons P(A|S=0, 1). This is clearly described in the results. Due to space constraints, the details of the tests are reported in the Supplementary Materials. We report modeling accuracy for $P(S| X_i)$, as well as describe the construction of the summaries for high-dimensionality features.
> >
> > - **“ It largely feels that the analysis of distribution shifts could be does by estimating the simple disease prevalence among subgroups or otherwise measuring the homogeneity of the dataset using something like optimal transport dataset distance”**
> >
> > Measuring disease prevalence amongst subgroups is not possible for EHR as we need to condition on multiple variables beforehand. For dermatology, this is essentially what the test does but it is a more generalizable approach as it can deal with multiple demographic attributes without reducing the power of the test (while you would be constrained by the number of cases in an intersection for every condition). Based on the causal graph, the conditioning needs to be performed (which we do with IPW). The statistical test could be different though, as we clearly mention, and as explored in Rabanser et al. This does not decrease the value of the proposed approach.
> >
> > The procedure and execution of the proposed statistical test is unclear
> > -----------------------------------------------------------------------------------------------
> >
> > - **“The procedures in setting up and evaluating these models are not made clear, nor are the underlying effects of possibly incorrect models on the overall ability of the statistical test to adequately reject the null hypothesis.”**
> >
> > We describe our testing strategy in detail in Appendix (A3) and in the Supplementary Materials for the medical applications. As mentioned in section 3 (line 103), Appendix A3.3 extensively investigates the tests, assessing its Type I error under different sample conditions.
> >
> > - **“Additionally, it's unclear whether Algorithm 1 is used on each variable in the BCN individually or all at once.”**
> >
> > Algorithm 1 is used on each variable separately, as described on line 93 (“For each variable U in the graph”) and in the table caption (“Independence and conditional independence testing to assess the nature of a shift S on a variable U ∈ G”). We have added a confirmation in the caption of algorithm 1:
> >
> > “Algorithm 1: (Conditional) independence testing assessing the nature of shift S on a *single* variable U ∈ G.”
> >
> > - **“Beyond this, it is difficult to adequately gauge the validity or helpfulness of the proposed statistical tool with the datasets used in the paper. Here, the design of synthetic examples where a subset of the variables are affected by the shift (rather than all of them) would be very insightful“**
> >
> > We describe briefly our validation strategy in section 3 (line 103), which points to synthetic experiments where we have performed this test and show that not only is our testing strategy performing well, our causal model assumptions seem to align with the data.
> >
> > Additional points where the clarity of the paper could be improved
> > -----------------------------------------------------------------------------------------
> >
> > - **The use of notation in Algorithm 1 is terse and not introduced anywhere. It's difficult to follow and understand. This is made harder by compounding the use of 'S' to be both the sample from the datasets and the source/target indicator.**
> >
> > We have amended the notation for clarity.
> >
> > - **It would be instructive to outline the BCN basic structure or concepts using a figure early on. This could be particularly useful in the design and execution of synthetic examples where the causal graph is modified accordingly.**
> >
> > BCNs are described in Appendix A.2.

---

> > > ### Author Response · Authors · 2022-08-01
> > > **Point by point response - part III**
> > >
> > > - **It's unclear what the training/testing procedure is to assess model fairness before and after the dataset shift. Fairness properties of a model depend significantly on the training mechanism of the model. The absence of any discussion along these lines is concerning.**
> > >
> > > These are briefly mentioned in the experiments, including the numbers of test cases in the source dataset. All training and validation parameters are described in detail in the Supplementary Materials, including hyper-parameters, and the use of a separate validation set. We note however that the overall generalization of the models from source to target is satisfactory, with the EHR model even performing better on the target than on the source. Therefore, the observed fairness gaps are not due to model overfitting, but to changes in the data distribution that directly affect fairness.
> > >
> > > - **In the EHR experiments, only 5 comorbidities were accounted for in the target population. How were these chosen? Why only those 5?**
> > >
> > > This is a misunderstanding. We consider 30 comorbidities, as displayed in Figure 3(c) and detailed in Appendix. We obtain statistical significance for 5 comorbidities out of the 30 when comparing the source and the target. This was clarified in the text.
> > >
> > > EHR experiments
> > > -----------------------------
> > > - **“It is stated that many hospitals do not have specialized ICU departments. This assertion is not supported by citation and seems speculative and misleading.”**
> > >
> > > This is not speculative, and we directly refer to the Critical Care Statistics (https://www.sccm.org/Communications/Critical-Care-Statistics , line 172). The Beth Israel medical system (where the MIMIC dataset is recorded from) is a special case and actually allows us to investigate the effects of training models on clinical subpopulations and applying them to others.
> > >
> > > - **“The poor model performance between the two subsets of data is not at all surprising since the subsets of the data cover very disparate patient populations.”**
> > >
> > > The model performs on par with the literature (performance of 78.6%, reported in Supplement). More importantly, the overall model performance is similar, if not higher for the target population than for the source. This means that the model is generalizable between the clinical subpopulations if performance is the metric considered. This statement is hence not correct.
> > >
> > > - **“The design of these experiments is not very intellectually honest without proper description or justification. [...] it appears that the authors are trying to construct a dramatic example to garner more significance. [...] there are some significant concerns about the intentions of the authors in using these experiments.”**
> > >
> > > We strongly disagree with those comments. First, this experiment was designed based on the input from clinicians. Second, we clearly mention the hypothesis we test (line 172), and how this is not a clinically surprising result (line 217). As mentioned in the text, the model does not have access to the ‘reason of visit’ for training, which is typical for such EHR models. As mentioned above, the model performs well on both populations overall and the difference is only in how the distribution shift affects fairness. Therefore, this comparison is not an “extreme case” or dishonest. We use this example to highlight how fairness mitigation in the source does not guarantee the transfer of fairness in the target (section 5).

---

> > > > ### Author Response · Authors · 2022-08-01
> > > > **Point by point response - part IV**
> > > >
> > > > Questions
> > > > ----------------
> > > > - **“Unfortunately, I am not sure whether the authors will be able to sufficiently address the major concerns that I raised in my review above that would allow me to raise my score without re-submission. There are core issues to the framing and narrative of the paper necessitating a full revision as well as some additional experimental analysis that would need to be re-reviewed before acceptance for publication. “**
> > > >
> > > > We have not identified any suggestions for modifications of the narrative, nor suggestions of additional experiments that have not already been performed (see Appendix and Supplement). Therefore, we kindly ask the reviewer to revise their assessment of our work.
> > > >
> > > > - **“In the authors’ view, what are the major intended contributions of this paper (see first major points in the section above)?”**
> > > >
> > > > As described in our point by point response, our main contribution is the testing approach.
> > > >
> > > > - **“How does this paper differentiate from Creager, et al (2020 ICML)? Is it only in the use of real-world medical datasets and the corresponding simple prediction tasks?”**
> > > >
> > > > The setup in Creager et al., 2020 is quite different from ours. First, the causal models are different, with Creager et al. 2020 investigating feedback loops in a system that might affect not only the demographic distribution, but also the decision-making process, i.e. X → Y. Shifts are also observed over multiple time points, rather than seeing a new dataset without adapting the model. Learning a policy is also quite different from predicting an output, leading to different mitigation strategies. Last but not least, our datasets represent real-world applications and include high-dimensional features, which the method proposed in Creager et al., 2020 cannot handle.
> > > >
> > > > - **“How sensitive is Algorithm 1 to the quality/accuracy of the models used internal to the statistical test?”**
> > > >
> > > > This is a valid technical point that we discuss (line 316). See Appendix A.3 for an investigation of the effect of the number of samples on test Type I error.
> > > >
> > > > - **“What efforts were taken in the construction of the experimental subpopulations to create a fair comparison of model performance?”**
> > > >
> > > > For all experiments, we consulted with clinicians and domain experts. We used state of the art model architectures. Our results are described in detail in the Supplement, highlighting the good performance of the model on the source, but also on the target populations.
> > > >
> > > > Limitations
> > > > ---------------
> > > >
> > > > - **"However, it is unclear what kinds of assumptions and limitations that exist for the development and execution of the statistical tests in Alg. 1. I can guess that the data needs to admit some form of conditional independence estimation (which might be a fairly strong assumption on its own) as well as be able to adequately estimate the dataset source (predicting)."**
> > > >
> > > > The limitation of the method is mostly that we need a causal graph of the application. This is clearly stated in the discussion (line 326).

---

> > > > > ### Comment · Reviewer_4fVY · 2022-08-03
> > > > > **Thank you**
> > > > >
> > > > > **General Response**
> > > > >
> > > > > Thank you for your patience with my laundry list of considerations, some of which derived from an incomplete understanding of the work. I admit that my comment on the ability of the authors to improve the paper sufficient for me to raise my score was misguided and incorrect. I apologize for the rash response. There are several issues or challenges with the writing and clarity however that should be addressed before I would be confident in recommending the paper for publication. I will try to highlight these comments in-line with the responses given by the authors.
> > > > >
> > > > > I evaluated the paper _as submitted_, the extensive justifications of unclear or missing information with reference to the Appendix or Supplementary material are helpful. I do however contend that the information surrounding the content, intent and, purpose of each portion of the referenced sections in the appendix is not always clear in the main paper. This left the impression of the paper being incomplete.
> > > > >
> > > > > Many of the recommendations/comments I left were in place in attempt to guide the improvement of the paper, despite my general dissatisfaction with the concept of "fairness transfer" under distribution shift as defined in this paper (it is not a tangible component of the model and is rather a byproduct of training + data limitations). It is important to be precise and clear with what the source of challenges are and what avenues for solution are being explored. Shorthand can be helpful if clearly and explicitly defined. This paper, as submitted, struggles to clearly outline the major contributions (as mentioned previously and graciously addressed by the authors) and mostly uses shorthand to discuss the general challenges and solution approaches at a high-level which hinders full understanding of the specifics of the proposed statistical model. In places, this could be mediated by a simple, single sentence explanation (specific examples given in-line with the comments and responses above).
> > > > >
> > > > >
> > > > > **Re: Creager, et al (2020) and other related work**
> > > > > Thanks for your explanation. It might be helpful to include this level of detail when discussing prior approaches to help isolate the contributions of the paper. In total, I find the related work discussions anchoring the conceptual framing of the paper to be uninformative. Understandably, the focus in Section 5 is on the mitigation strategies and performance comparisons. However, it is difficult to make comparisons as the performance metrics are not easily accessed in the paper. Perhaps summarizing the most relevant comparisons in a table would be helpful?
> > > > >
> > > > > **Sensitivity of Alg. 1**
> > > > > This insight is not apparent from Section A.3. The metrics and procedures used to evaluate the proposed approach are not clearly outlined. I presume that the authors are referring to the analysis underlying Figure 3c. This specific experiment is not sufficiently introduced or outlined. It would be helpful to introduce the procedures in Section A.3.3 more directly (e.g. "We investigate the effect of domain shift on the observed R.V.s using the proposed statistical test. We evaluate using the proportion of false positives, which indicate that..... We also investigate the effect of the number of samples when estimating the domain of the dataset, a critical component of our proposed statistical test..."). As someone who is not entirely familiar with the material or type of statistical tests performed in this paper, taking a little more care in discussing what I'm expecting to look at would go a long way in improving the overall clarity of the paper.
> > > > >
> > > > > **Design of cohorts**
> > > > > The paper, as submitted, is devoid of any information that discusses or explains this. _This is a critical omission._ Without full transparency of the motivations underlying the design of the experiments and the supporting cohorts (including the clinical guidance that directed these efforts), it is very easy to question the intention of the experiments. A summary of this information *needs* to be included in the main body of the paper.

---

> > > > > > ### Author Response · Authors · 2022-08-03
> > > > > > **Discussion**
> > > > > >
> > > > > > We would like to thank the reviewer for engaging in the discussion and really appreciate their re-assessment of our work.
> > > > > >
> > > > > > We hope that our further changes to the text answer your concerns (see below, in our other responses to specific points, and in the uploaded revised paper).
> > > > > >
> > > > > > **Sensitivity of algorithm 1**
> > > > > >
> > > > > > We have amended the text in section 3 to be more explicit about the experiments performed in the Appendix:
> > > > > >
> > > > > > *"We conduct three validation experiments for our testing procedure using data from our dermatology application in Sec. A.3.3. We first assess Type I error in an experiment where we test random splits of the same data against each other. Our test displays a false positive rate similar to our threshold of 5% for hypothesis testing. We however note that the variance of this result increases with smaller numbers of samples. We then confirm non-trivial power in an experiment where we introduce a shift by subsampling younger patients with a particular skin condition in the source dataset (a shift on $A$ and $Y$). Our testing approach correctly identifies the dimensions of $A$ and $Y$ that were affected by $S$ due to the engineered shift. We replicate this test for a shift in $Y$ only, with the same result."*
> > > > > >
> > > > > > We have also modified the text in Appendix A.3.3 to be more explicit:
> > > > > >
> > > > > > *"We assess the specificity and sensitivity of our testing procedure (see Algorithm 1) using engineered shifts on the dermatology data. To estimate Type I error, we compare the source data to itself, which should lead to all variables being independent of the environment. More specifically, we select $10,000$ random samples from the dataset to compute the weights, and another $1,000$ random samples to apply the weights on and perform the weighted test. These sets are boostrapped 100 times (separately for the source and target sets) and we assess the proportion of false positives that is obtained for each variable $U$. We expect that we will obtain ~5% of false positives, as defined by our hypothesis testing threshold. As we do not expect any relationship between  \{$\mathbf V$\} and $S$, we use a logistic regression to estimate $P(S = s \mid \mathbf{V})$.'*
> > > > > >
> > > > > > [...]
> > > > > >
> > > > > > *"As the number of samples used to compute the weights is an important component of our testing approach, we repeat the process while varying the number of samples from 100 to 5,000. Figure 5(c) displays that the variance on Type I error (here across conditions) decreases when the number of samples to fit the $P(S = s \mid \mathbf{V})$ classifier increases."
> > > > > >
> > > > > > [...]
> > > > > >
> > > > > > For sensitivity: *"Therefore, the testing approaches produces a causal graph that mirrors the graph used to engineer the shift, and our procedure is considered faithful."*
> > > > > >
> > > > > > **Design of cohorts**
> > > > > >
> > > > > > We have added relevant details for the EHR experiments (see the corresponding response).
> > > > > >
> > > > > > **Related work**
> > > > > >
> > > > > > We have added a sentence about Creager et al. (2020) in section 5 (fairness mitigation):
> > > > > >
> > > > > > *"More similar to our settings, the work by \citet{Creager2020} addresses fairness in dynamical systems, although only for low-dimensionality variables."*
> > > > > >
> > > > > > Similarly, we have added a sentence about the work referred by Rev. aaBT in the "shift detection" subsection. We believe this section provides most of the context for the development of the tests.

---

> > > > ### Comment · Reviewer_4fVY · 2022-08-03
> > > > **Re: EHR experiments**
> > > >
> > > > Clarifying and adding additional detail about the cohort selection, experiment design and underlying motivations will greatly improve the paper in this section. It is critical that this is included in the main body of the paper. I apologize for misunderstanding the [noa] reference. Perhaps greater clarity of where that citation fits into the discussion could help? By amending that sentence to say, "As reported by the Critical Care Statistics [noa], over half of the hospitals in the US...".
> > > >
> > > > This would then set up a much clearer introduction to the use of MIMIC in this paper. Something along the lines of: "However, BIDMC is unique in that it has several ICU departments. This enables an exploration into the possible dangers of model performance under distribution shift allowing us to investigate the effects of training models on clinical subpopulations and applying them to others." --> More transparency about the design and motivation of these experiments is the best choice here.
> > > >
> > > > Also, while the model doesn't have access to "reason for visit", it is a clear confounder to the health presentation of patients. Elective cardiac surgery patients are typically in better health, skew slightly younger (and male), and have shorter stays. Other underlying conditions such as hypotension are more prevalent in surgical ICUs due to blood loss incurred during the procedures. Without correcting for this bias, possible comparisons in model performance may not be valid. It is possible that the proposed IPW component of the proposed statistical test helps account for this, it is however an outlying consideration that goes unmentioned in the cohort selection of this part of the empirical results.

---

> > > > > ### Author Response · Authors · 2022-08-03
> > > > > **EHR experiments amended writing**
> > > > >
> > > > > Thank you for these suggestions. We have amended the writing accordingly and have added details on the cohort selection:
> > > > >
> > > > > *"We use the open access, de-identified Medical Information Mart for Intensive Care III (MIMIC-III) dataset \citep{Johnson2016-eu}, which consists of data from admissions to ICUs at the Beth Israel Deaconess Medical Center between 2001 and 2012. This clinical system has the benefit of having separate specialized ICUs and allows to assess the generalizability of risk scores estimated in one clinical population to another. Based on clinical input, we consider the Medical ICU (MICU), Surgical ICU (SICU) and Trauma Surgical ICU (TSICU) to be generalist ICUs (source data); whereas the Cardiac Surgery Recovery Unit (CSRU) and Coronary Care Unit (CCU) are specialized ICUs (target data). This split leads to 17,641 patients included in the source dataset and 10,442 patients in the target dataset, after selecting adult patients with a length of stay of minimum one day and with a recorded care unit. Our goal is to obtain a robustly fair model that predicts prolonged ICU stay (i.e. length of stay $>3$ days, as in \citep{Wang2019-ac,Pfohl2021-xn}) using data from the first 24 hours of first ICU admission and a recurrent architecture \citep{Tomasev2019-tu,Tomasev2021-uf,Roy2021-rr}. The model is trained on 80% of the source data, tuned using a separate split of 10% and tested on the remaining 10%. See the Supplement for further details."*
> > > > >
> > > > > Regarding the confounding "reason for visit": we agree with the reviewer and this is stated in the results of the EHR section. The IPW potentially (partly) corrects for this based on comorbidities and treatments. Our results suggest that there is no significant differences in length of stay after correcting for these factors, although the high-dimensionality of the correction can lead to under-powered results (see the Supplement for a discussion of these specific results). We have added this explicitely below the section above:
> > > > >
> > > > > *"Noteworthy, the model does not have access to the `reason for visit'."*

---

> > > ### Comment · Reviewer_4fVY · 2022-08-03
> > > **Thanks for addressing and clarifying these things**
> > >
> > > Thanks for the detailed response. It has clarified my thinking underlying each of my comments.
> > >
> > > Quickly re: Nestor, et al... I wanted to quickly mention that I drew favorable connections between the cross sectional analyses presented in each work in the presence of distribution shift. The intention of the analyses differs between the work (not in any way to diminish the expected contributions of this paper) but could fall under the same model generalization under transfer umbrella, if you will.

---

> > ### Comment · Reviewer_4fVY · 2022-08-03
> > **Problems can be addressed with clearer and more explicit writing**
> >
> > Thanks for addressing the concerns about omitted literature. It has helped me better frame the contributions of this work.
> >
> > I understand that there is repeated use of the statistical test throughout the paper. My comment here derives from how the paper is written, at times (due to the heavy reliance of the appendix to demonstrate the use of Alg. 1) the statistical test which is claimed to be the major contribution of this work alongside the causal framing seems to be an afterthought. I recommend that more explicit references and clearer organization (perhaps through a table comparing results, elucidating comparisons with bolded indications of discovered factors to fairness properties of the models failing to be maintained across the distribution shift) would help improve this aspect of the paper.
> >
> > **Re: mirroring Chen, et al (2021):** In my estimation, the categorization used as well as some of the recommendations are directly derived from the original paper. The proposed adjustment to the language preceding these recommendations is great. I believe that this type of attribution is more appropriate than what is currently written in the paper where a reader may be inclined to assume that this categorization is wholly original to this work.
> >
> > **Re: "fairness transfer" framing:** From the very first section (lines 28-30) there are mentions of "problematic aspects of the data". As stated in my review, the challenges come through possible insufficiencies in the source or target data to support model generalization, especially when considering cross-sectional performance as analyzed in this work. I think more concrete discussion of the interplay between training and the properties ingrained in the model as a response to characteristics/limitations of the underlying data (possibly acknowledging unobserved confounding) would be appreciated.

---

> > > ### Author Response · Authors · 2022-08-03
> > > **Amending the writing**
> > >
> > > Dear Reviewer,
> > >
> > > Thank you for clarifying your comments and making suggestions.
> > >
> > > **Use of the testing approach**
> > >
> > > We have now clarified the writing with respect to the tests, highlighting that these are used throughout the experiments. For instance:
> > >
> > > Section 4.1: To understand the failure of fairness transfer in this context, we investigate the structure of the shift *by referring to Algorithm 1.*
> > >
> > > Section 4.2: We now test whether the shift $S$ has direct effects on the variables in this problem *using our proposed approach.*
> > >
> > > We believe that our figures 2(a) and 3(a) serve the purpose of illustrating the results of our testing approach, as we add (colored) links to the causal graphs based on the results of each test (as described in the text). We have amended the caption of these figures to be more explicit:
> > >
> > > *"Colored arrows represent statistical dependence as identified by Algorithm 1."*
> > >
> > > **fairness transfer framing**
> > >
> > > Thank you for clarifying your comment, we know understand how this can suggest a "data only" framing of the issue. In the text, "problematic" means that the variables are affected by the shift in such a way that fairness transfer cannot be guaranteed. As we mention in the discussion, not identifying an issue is however not a guarantee that fairness will transfer. We have clarified this statement and added a (brief due to space constraints) discussion of the broader issues:
> > >
> > > line 28-30: ...*identify aspects of the data that hinder the transfer of fairness properties*
> > >
> > > discussion (algorithm development subsection): *"We however note that our work does not assess the impact of model architecture or different training strategies, which might lead to compounded bias and/or poor generalization."*
> > >
> > > We acknowledge unobserved variables in their general sense in the discussion, based on the suggestion of Rev. aaBT.

---

> ### Author Response · Authors · 2022-08-08
> **Follow up**
>
> Dear Reviewer,
>
> Thank you again for acknowleding our response and clarifying some of your concerns.
>
> We hope we have addressed your comments in our point by point responses in each part. If so, we would appreciate it if you would consider revising your score or let us know if there are still remaining issues unaddressed so that we can respond to them during the discussion.
>
> Thank you,
> The authors

---

> > ### Comment · Reviewer_4fVY · 2022-08-08
> > **re: follow up**
> >
> > Hi, thanks for following up!
> >
> > I appreciate your responses and the insight you've been able to provide. I'm happy to see that the paper has been improved, most importantly its clarity, as it has helped me to re-evaluate the work and view it more positively. I feel well informed ahead of the reviewer discussion.

---

### Author Response · Authors · 2022-08-01
**Response to reviewers**

We thank the reviewers for their comments and suggestions. We provide here a summary of our response to the reviewers and of the changes made in the manuscript:

- We have added 2 references that were suggested by reviewers aaBt and 4fVY and have in particular discussed the recent work of Singh et al., 2022 that is directly relevant.
- We have modified the notation and added details when this led to misunderstandings.
- We have added to our discussion of the limitations of the method in terms of handling multiple environments or with regards to unobserved demographic variables (Reviewer aaBt).
- We have clarified that our work includes multiple experiments to validate our proposed testing approach. Due to space constraints, these have been included in the Appendix and detailed results are presented in the Supplementary. (Reviewer 4fVY)
- We have clarified that all our experiments were designed with input from clinicians, that our models reflect state-of-the-art performance and that all details of the modeling can be found in the Supplementary Materials. (Reviewer 4fVY)
- We have added another synthetic experiment as suggested by Reviewer Ezbv (Appendix).

We hope this addresses your concerns and clarifies any misunderstandings. We would be happy to make further changes or respond to more questions if any.

---

### Comment · Area_Chair_qySw · 2022-08-08
**Let's wrap up an active discussion phase!**

Hi folks --

Thanks to the reviewers for their initial reviews, to the authors for a thoughtful rebuttal, and to both sides for their discussion in the meantime.  Reviewers -- is there anything else that would be helpful to ask the authors before we move to our final deliberative phase?  Please do get those final comments posted!

-- AC

---

### Meta-Review · Area_Chair_qySw · 2022-08-29

**Recommendation:** Accept
**Confidence:** Certain

**Metareview:**

This is a compelling work characterizing some forms of model (non-)robustness to drift through a causal lens, with a focus on performance metrics including group-level fairness.  The methodological novelty to the work is a method for discovering structure for that drift, then using that structure to (i) estimate impact on metrics and (ii) mitigate those impacts.  That method requires a domain expert to provide a rough causal graph for the application at hand, which is a light negative; yet, the work then presents two in-depth studies in the healthcare space to argue that this is a surmountable requirement, at least in some settings.  This is a tough space to operate in, and I appreciated this very deep dive into two "real" cases -- as did the reviewers.

**Award:**

No

---

### Decision · Program_Chairs · 2022-09-14

Accept